# FIPER: Factorized Features for Robust Image Super-Resolution and Compression

**Yang-Che Sun**[1]    **Cheng Yu Yeo**[1]    **Ernie Chu**[2]    **Jun-Cheng Chen**[3]    **Yu-Lun Liu**[1]

[1]National Yang Ming Chiao Tung University    [2]Johns Hopkins University    [3]Academia Sinica

## Abstract

In this work, we propose using a unified representation, termed **Factorized Features**, for low-level vision tasks, where we test on **Single Image Super-Resolution (SISR)** and **Image Compression**. Motivated by the shared principles between these tasks, they require recovering and preserving fine image details, whether by enhancing resolution for SISR or reconstructing compressed data for Image Compression. Unlike previous methods that mainly focus on network architecture, our proposed approach utilizes a basis-coefficient decomposition as well as an explicit formulation of frequencies to capture structural components and multi-scale visual features in images, which addresses the core challenges of both tasks. We replace the representation of prior models from simple feature maps with Factorized Features to validate the potential for broad generalizability. In addition, we further optimize the compression pipeline by leveraging the mergeable-basis property of our Factorized Features, which consolidates shared structures on multi-frame compression. Extensive experiments show that our unified representation delivers state-of-the-art performance, achieving an average relative improvement of 204.4% in PSNR over the baseline in Super-Resolution (SR) and 9.35% BD-rate reduction in Image Compression compared to the previous SOTA. Project page: https://jayisaking.github.io/FIPER/

## 1 Introduction

Single-image super-resolution (SISR) aims to recover high-quality images from low-resolution inputs, with accuracy depending on precise restoration of fine details and geometric correspondences (e.g., stripes, grids, textures). CNN pioneers [24, 59] were followed by GAN-based methods for perceptual realism [47] and Transformer networks for long-range context [14]. Swin-based variants further boosted performance [22, 18, 19, 71], inspiring ever-heavier architectures. These prior works optimize network design while neglecting the underlying image-content representations themselves. On the other hand, decomposition-based methods[66, 44, 39] seek to model the frequency components directly through adding Fourier- or Wavelet-like bottlenecks in the middle. Although these designs can indeed better capture the recurring visual patterns in learning effective SR representations than aforementationed methods, as illustrated in fig. 1, they still suffer from poor perceptual quality.

This raises a critical question: beyond a simple network output, can we derive a formulation that more effectively captures these patterns and aligns with the goals of SISR?

Conventional multi-scale pyramids aggregate features at several resolutions, but they store all activations densely, do not disentangle frequency content, and cannot be shared directly across tasks. Instead, our factorized descriptor decomposes images into generalizable basis $\times$ coefficient pairs, providing an explicit multi-frequency handle through different modulations.

On the other hand, image compression serves as a fundamental task in low-level vision applications, where the traditional compression standards [41, 95, 92] lay the groundwork. The emerging learned

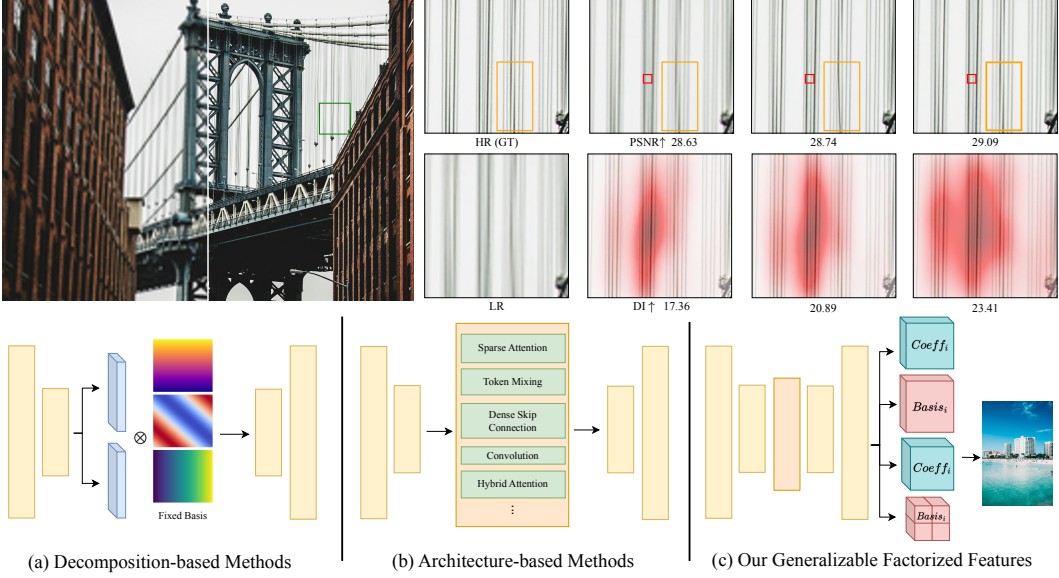

Figure 1: In this work, we propose to represent images by **Factorized Features**, contrasting with prior methods that either (a) rely on fixed-basis decomposition[66, 39] or (b) introduce various complex model blocks [15, 71] such as sparse attention, dense skip connection, etc. As highlighted in the **orange** boxes, (a) successfully captures periodic line structures but suffers from lower fidelity, while (b) cannot recover those repeating patterns without explicit frequency modeling. In contrast, our method (c) uses learned decomposition with generalizable bases to deliver high-fidelity reconstruction while preserving periodic structures. We also present LAM [32] and Diffusion Index (DI) [32] for reconstructing the **red**-boxed patch, with its pixel-importance map shown on the middle row; a higher DI indicates more pixel utilization.

image compression models [8, 3, 34, 62, 80, 20], compression algorithms of which mostly follow the pixel-space transform coding [8, 31] paradigm, then introduce a variety of networks [96, 62, 106, 67, 27] or difference losses [35] to further optimize compression efficiency by learning more compact latent representations and improving reconstruction quality. Specifically, they convert pixels into compact representations through a transform module, which eliminates the redundancy and reduces the bit cost in the subsequent entropy coding process. However, the core challenge of image compression is to accurately reconstruct the information lost during compression and quantization. In other words, the models can essentially be viewed as reconstructing a high-quality image from its 'low-resolution' version, much like Super-Resolution or all other common low-level vision tasks such as image restoration, deblurring, denoising, etc.

Based on the aforementioned analysis, it becomes clear that although these tasks appear to be distinct, they share mutual similarities in two key aspects: (1) The tasks require models to restore fine details from low-quality image content, as well as implicitly capture and reconstruct repetitive structural elements. (2) They aim to conserve image quality, either by enhancing resolution or efficiently compressing data without significant loss of perceptual fidelity. Hence, inspired by recent advances in decomposition fields and matrices factorization in 3D scene modeling [11, 81, 9, 28, 7, 12, 30], we propose a unified representation, **Factorized Features**, with generalizable Coefficient Backbone and Basis Transformer for learned coefficient and basis, respectively. This approach explicitly captures multi-scale visual features and repetitive structural components in images through a basis-coefficient decomposition. The resulting representation strikes a balance between being compact and information-rich, enabling the resolution of structural ambiguities and the precise modeling of image details through a multi-frequency formulation.

Finally, we also propose consolidating multiple bases into one through a network for multi-image processing, as in traditional Discrete Fourier Transform, all signals of same size are reconstruct by the same set of basis functions, and we argue that learned shared basis can leverage the mutual information across multiple images to capture common structures.

The main contributions of this paper are summarized as follows:

- We identify the weakness of previous similar representations and propose a refined representation, Factorized Features, based on thorough analysis as in section 4.1 for coarse and fine details of images via explicitly modeling multi-scale visual features and structural components.
- We generalize such image descriptors to learned settings on super-resolution and image compression, with potential applicability to broader low-level vision tasks.
- We demonstrate state-of-the-art performance on benchmarks for both super-resolution and image compression through extensive experiments.

## 2  Related Works

**Super-Resolution (SR).** Image super-resolution (SISR) aims to reconstruct high-resolution (HR) images from low-resolution (LR) inputs, which is crucial for computer vision. Early CNN-based methods [24, 59, 113, 89, 43] introduced residual [48, 61, 94, 112, 111, 21, 58] and recursive learning [19, 90]. GAN-based approaches [47, 48, 97, 98] improved perceptual quality but faced spatial-locality constraints, leading to Transformer-based SISR models [14, 53]. SwinIR [55] integrated window attention with Swin Transformer [69], inspiring further advancements [22, 114, 109, 21, 71]. Hybrid models such as CRAFT [50], DAT [18], and HAT [15] optimize feature aggregation, while RGT [19] enhances spatial details efficiently. Recent efficient Transformer variants such as Restormer [105] and Uformer [99] demonstrate that lightweight attention or activation-free blocks can match or surpass heavier designs. Diffusion-based methods such as DDRM [42] provide an orthogonal line of plug-and-play restoration priors, with recent advances extending to video restoration [104], reference-based face restoration [37], and inpainting tasks [64]. Additionally, learned pipeline approaches [68] reverse the image formation process for reconstruction tasks.

**Spectral Coordinate Transforms.** A separate line of work seeks to encode coordinates rather than deepen the backbone. Fourier Features [91, 78] map input positions to a fixed harmonic basis, enabling MLPs to model high-frequency detail. CoordConv [65] expands convolutional layers with raw $(x, y)$ channels to facilitate spatial reasoning. On the other hand, sawtooth coordinate transformation $\gamma(\cdot)$, of our Factorized Features, *folds* space into a piece-wise linear triangle wave and produces evenly spaced spectral peaks (see section 4.1) that favor sparse and low-rank Factorized Features. Unlike the above encoding, our basis-coefficient decomposition paired with $\gamma$ can be trained end-to-end and reused directly in both super-resolution and compression pipelines.

**Image Compression (IC).** Deep learning surpasses traditional codecs such as JPEG [95] and JPEG-2000 [92], with CNN-based [8, 4, 33], Transformer-based [45, 115, 100, 25, 27, 106, 67, 29], and GAN-based [73, 74, 26, 86] approaches improving performance. ELIC [36] introduced adaptive coding; LIC TCM [62] combines CNNs and Transformers; eContextformer [46] leverages spatial–channel attention. GroupedMixer [51] introduces token mixers, while Wavelet Conditional Diffusion [88] balances perceptual quality and distortion. Generative codecs such as HiFiC [77] pursue high-fidelity reconstructions at extremely low bitrates and offer a complementary direction to likelihood-based models.

Our work diverges from these trends by introducing Factorized Features, a unified framework modeling both visual and structural features to enhance performance across super-resolution and image compression.

## 3  Frequency Decomposition Preliminary

In this section, we start by revisiting several previous renowned representations for images, e.g., 2D Fourier Transform, Learnable Fourier Series [66], and Factor Fields [12], and then generalize them to a concise formulation of basis-coefficient decomposition as in eq. (3).

The Fourier Transform has long been used in many scenarios to represent periodic or finite signals with a group of sinusoidal functions. In practice, it often takes the form of the Discrete Cosine Transform for a real-valued 2D discrete signal $f$ with height $H$ and width $W$:

$$f[x, y] = \sum_{u=0}^{H-1} \sum_{v=0}^{W-1} S_{u,v} \cos\left[\frac{\pi u}{H}\left(x + \frac{1}{2}\right)\right] \cos\left[\frac{\pi v}{W}\left(y + \frac{1}{2}\right)\right], \qquad (1)$$

where $S_{u,v}$ is the amplitude.

Without the loss of generality, we consider square images in this work, with edge length (and also the period) denoted as $T$:

$$f[x, y] = \sum_{u=0}^{T-1} \sum_{v=0}^{T-1} S_{u,v} \cos \left[ \frac{2\pi}{T}(ux + vy) - \xi_{u,v} \right],$$ (2)

where $x < T$, $y < T$, and $\xi_{u,v}$ is phase shift.

In image processing tasks, the goal is to learn $S$ and $\xi$, then use the formula above to reconstruct high-quality images from degraded inputs. To achieve this, [66] treats $S$ and $\xi$ as learnable parameters, allowing the model to optimize them for better reconstruction. Furthermore, the frequencies $u$ and $v$ are also designed to be promptable by the model, enabling dynamic adaptation to spatial-varying high- and low-frequency components.

More generally, we can formulate it as the multiplication of coefficient $c$ and basis $b$ with simplified notation, and in this work, we use $\mathbf{x}$ to represent the pixel coordinates $(x, y)$:

$$f(\mathbf{x}) = \sum_{i=1}^{N} c_i \cdot b_i(\mathbf{x}), \; b_i(\mathbf{x}) = \cos \left[ (u_i x + v_i y) - \xi_i \right],$$ (3)

where $N$ denotes the number of frequency components.

In the next section, starting from the concise formulation in eq. (3), we introduce our Factorized Features framework and elaborate on its spatially variant coefficients and adaptive basis functions, while showing its application to super-resolution and image compression.

## 4 Methods

In this section, we dive into the motivation and how we resolve the weaknesses of previous decomposition-based methods by our Factorized Features in section 4.1. Next, we discuss how to integrate such a strategy into learned settings of Super-Resolution in section 4.2 and Image Compression section 4.3. We provide the background of learned image compression in the supplementary materials.

### 4.1 Formulation of Factorized Features

In this subsection, we focus on the analysis of the properties from different factorization components, and derive our final Factorized Features through these insights, starting from the simplest expression eq. (3).

**Spatially Variant Coefficient.** Intuitively, the local frequency spectrum of a signal is spatially variant, meaning that different regions exhibit distinct spectral characteristics, e.g., certain regions of an image do not require high-frequency components. Therefore, in this work, we set coefficient $c_i$ to be spatial-varying, following [12], i.e.,

$$f(\mathbf{x}) = \sum_{i=1}^{N} c_i(\mathbf{x}) \cdot b_i(\mathbf{x}).$$ (4)

**Learned Non-uniform Basis.** Previous works[66, 54, 49] learn basis through frequencies $u, v$ and phase shift $\xi$; however, constructing signals with definitive sets of uniform sinusoidal harmonic functions does not facilitate the utmost details. The predefined nature of these sinusoidal functions restricts local flexibility, making fine-grained adjustments more challenging. Also, since low-frequency components typically require more energy [52], during the learning process with training losses of MAE or MSE, models would attend more to low-frequency parts and sacrifice local details. As shown in fig. 2a, higher frequency components converge after their counterpart. Hence, in this work, we generate the entire basis map $b_i = M_i \in R^{T \times T}$ for better fitting and local reconstruction.

To this end, we have introduced our rough idea: Under the generalizable setting, we use networks to generate a multi-channel coefficient map and a multi-channel basis map, and combine them. In this work, the composition process happens in the end with a projection function $\mathcal{P}$ for RGB images $\hat{I}$:

$$\hat{I}(\mathbf{x}) = \mathcal{P}\left( \mathbf{Concat}_{i=1}^{N} \left\{ c_i(\mathbf{x}) \cdot b_i(\mathbf{x}) \right\} \right).$$ (5)

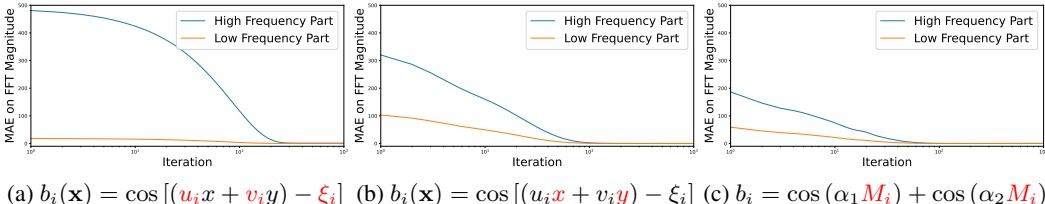

(a) $b_i(\mathbf{x}) = \cos\left[(u_i x + v_i y) - \xi_i\right]$ (b) $b_i(\mathbf{x}) = \cos\left[(u_i x + v_i y) - \xi_i\right]$ (c) $b_i = \cos(\alpha_1 M_i) + \cos(\alpha_2 M_i)$

Figure 2: **Comparison of different trainable basis settings** We test on single-image regression with $f(\mathbf{x}) = \sum_{i=1}^{N} c_i(\mathbf{x}) b_i(\mathbf{x})$ with equal parameter count. (a) fixed sinusoidal basis favoring low–frequency content, eq. (3); (b) learnable (i.e. requires gradient) coordinate mapping with moderate gains; (c) **Learned Non-uniform Basis** $M_i \in \mathbb{R}^{T \times T}$ converges more synchronously. Note that $\alpha_1, \alpha_2$ are the frequencies, functionally similar to $u, v$.

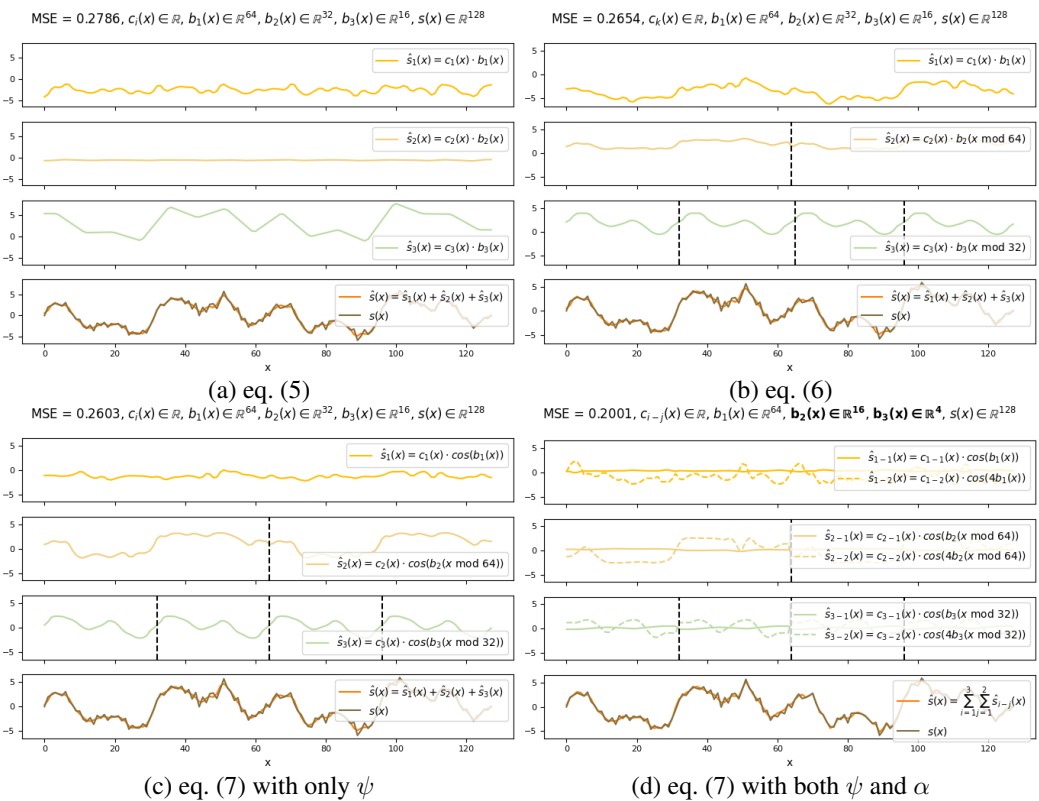

(a) eq. (5)  (b) eq. (6)

(c) eq. (7) with only $\psi$  (d) eq. (7) with both $\psi$ and $\alpha$

Figure 3: **Visualization of different factorization methods** This figure shows the difference between vanilla fields, coordinate transformation, and multi-frequency modulation. Specifically, (a) is a vanilla basis-coefficient field, and (b) adds sawtooth transformation. We can see in (b) that the coordinate transformation explicitly models a patch-like pattern, e.g., the second plot from the top is divided into two periods. By enforcing such frequency components, the models can decompose the signal effectively. Next, (d) is our full formulation, while (c) has no $\alpha$. Intuitively, from (b) to (c), applying $\psi$ (we use cos here) should not have much effect on the performance since we do not have any constraint on these learnable bases. With the introduction of $\alpha$, the models are forced to attend to signal components of different frequencies.

In contrast to previous works which employ Fourier bottlenecks[66], or frequency decomposition blocks[54, 49, 52] among model blocks, here we seek to address a fundamental question: Can a well-learned representation effectively replace the simple feature map output? That is, our solution is meant to be applied directly on networks to represent **any** images.

Next, we discuss more improvements on frequency decomposition to model image details.

**Coordinate Transformation.** The success of the Fourier-based representation originates from its sinusoidal decomposition. By using such periodic functions, one can effectively analyze and

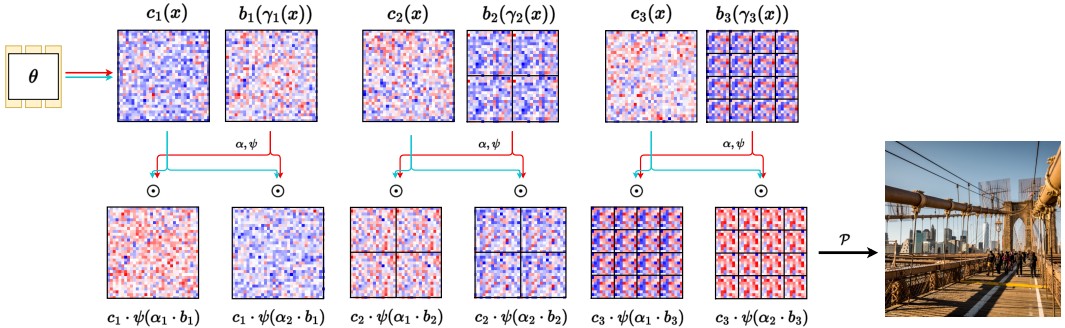

Figure 4: **The Proposed Factorized Features.** In this figure $\alpha = \{1, 16\}$, $\psi = \sin(\cdot)$, and that $\gamma_i(x)$ is coordinate transformation. We can see in this figure with sawtooth transformation $\gamma$ the feature is prompted to learn pattern from different scale in a patch-like manner, e.g. $\gamma_2$ makes the same basis repeat four times in the grid. However, only $\gamma$ is not enough for very fine details; therefore, we introduce $\alpha$ and $\psi$. Each basis is forced to accommodate both high-frequency (large $\alpha$) and low-frequency (small $\alpha$) components together with periodic functions $\psi$, i.e., larger $\alpha$ will make $\psi$ oscillate sharper. With explicit formulation of repetitive patterns and high frequency components, images can be represented with fine-grained details, which is especially useful for low-level vision tasks such as Super-Resolution and Image Compression.

reconstruct signals with the enforcement of high- and low-frequency components. However, the formulation in eq. (5) does not have such explicit harmonic feature (coefficient and basis are learned maps), and thus as in fig. 3((a) vs (b)) we witness performance loss due to lack of such guidance. Inspired by position encoding [78, 11, 81, 7, 9] and Fourier feature mapping [91], we introduce coordinate transformation function $\gamma_i$ on $\mathbf{x}$ before sampling basis $b_i$:

$$\hat{I}(x) = \mathcal{P}\Big(\mathbf{Concat}_{i=1}^{N}\Big\{c_i(x) \cdot b_i(\gamma_i(x))\Big\}\Big). \tag{6}$$

In practice, we empirically[12] use sawtooth transformation $\gamma(x) = x \bmod k, k \in \mathbb{R}$; visualization is also provided in the supplementary material. We argue that the function explicitly imposes patch-like periodic constraints and, therefore, enforces models to learn such repetitive patterns.

**Multi-frequency Modulation.** [85] has shown that neural networks are biased towards learning low-frequency content. Hence, we propose multi-frequency modulation to compel the model to fit high-frequency components, visualized in fig. 3c and fig. 3d:

$$\hat{I}(x) = \mathcal{P}\Big(\mathbf{Concat}_{i=1\,j=1}^{N\quad K}\Big\{c_{ij}(x) \odot \psi(\alpha_j \cdot b_i(\gamma_i(x)))\Big\}\Big), \tag{7}$$

where $\psi \in \sin, \cos$ and $\alpha \in R$. In this way, each basis $b_i$ contributes to both high- and low-frequency components by different values of $\alpha$. Consequently, if a basis is only accurate in the low-frequency domain, the output will contain undesired high-frequency noise.

Alternatively, eq. (7) can be interpreted as making the spatial coordinates $x, y$ of $b_i(\mathbf{x}) = \cos[(u_i x + v_i y) - \xi_i]$ in eq. (3) learnable instead of the frequency components $u, v$. To validate this, we conduct experiments (fig. 2b) showing that learning coordinates mitigates the rigid constraint of uniform distribution, aligning better with the non-uniform nature of visual patterns across spatial layouts; that is, some pixels share similar RGB values, indicating smooth color transitions, while others exhibit abrupt changes, suggesting high-frequency variations.

To this end, our full Factorized Features representation is formulated in eq. (7), visualized in fig. 4.

## 4.2 Super-Resolution with Factorized Features

We represent a super-resolved image using our Factorized Features, where coefficients and basis are generated by networks $F_{\mathbf{coeff}}$ and $F_{\mathbf{basis}}$ from a low-resolution Image:

$$\hat{I}_{\mathrm{SR}}(x) = \mathcal{P}\Big(\mathbf{Concat}_{i=1\,j=1}^{N\quad K}\Big\{c_{ij}^{\mathrm{LR}}(x) \odot \psi(\alpha_j \cdot b_i^{\mathrm{LR}}(\gamma_i(x)))\Big\}\Big), \tag{8}$$

where $c^{\mathrm{LR}}(x) = F_{\mathbf{coeff}}(I_{\mathrm{LR}})(x)$ and $b^{\mathrm{LR}}(x) = F_{\mathbf{basis}}(F_{\mathbf{coeff}}(I_{\mathrm{LR}}))(\gamma(x))$. Note that we sample the outputs $F_{\mathbf{coeff}}(I_{\mathrm{LR}})$ and $F_{\mathbf{basis}}(F_{\mathbf{coeff}}(I_{\mathrm{LR}}))$ with coordinates $x$ and $\gamma(x)$, respectively. We omit convolution layers here for simiplicity.

Figure 5: **Super-Resolution and Image Compression with Factorized Features.** This figure illustrates how Factorized Features are used in Super-Resolution and Image Compression. (a) Given a low-resolution input image, we first extract $X_{coeff}$ feature with Coefficient Backbone. Next, we generate the basis and coefficient with Basis Swin Transformer and convolution layers, respectively, from the same $X_{coeff}$. Finally, the prediction is reconstructed by Factorized Features Reconstruction. (b) To decrease distortion in image compression, we replace the synthesis transform of the traditional learned image compression pipeline with (a) by aligning spatial resolution and latent channels.

Our model comprises three main components: Coefficient Backbone $F_{\mathbf{coeff}}$, Basis Swin Transformer $F_{\mathbf{basis}}$, and Factorized Features Reconstruction. As shown in fig. 5a, the process begins with $I_{LR} \in \mathbb{R}^{3 \times H \times W}$. The Coefficient Backbone extracts features $X_{\mathrm{coeff}} \in \mathbb{R}^{C_c \times H_c \times W_c}$, which are then used to generate coefficients $c$ through convolution and pixel shuffle operations. Also, $X_{\mathrm{coeff}}$ is fed into the Basis Swin Transformer to produce a multi-scale basis $b = \{b_1, ..., b_N\}$, $b_i \in \mathbb{R}^{C_{b_i} \times H_{b_i} \times W_{b_i}}$. The coefficients and basis are combined to reconstruct $I_{SR} \in \mathbb{R}^{3 \times sH \times sW}$ using eq. (8), where $s$ is the scale factor.

We optimize model parameters using $L_1$ loss. To be more clear, when the Basis Swin Transformer is optimized, the basis is optimized simultaneously, i.e. the models are trained to adapt to a content-aware basis, not calculated by heuristic rules. This design enables content-aware basis generation: smooth regions get low-frequency bases while textured areas receive high-frequency ones.

To demonstrate the effectiveness of our method, we use existing SR methods [16, 18, 109] as the Coefficient Backbone. For the Basis Swin Transformer, we employ simple Swin Transformer Blocks [70] with a series of special downsampling operations. Specifically, we use a dilation-like downsampling technique to accommodate the $\gamma$ sawtooth sampling pattern, where the details and visualization can be found in Supplementary Materials. The final basis is refined using additional upsampling and convolution layers.

## 4.3 Image Compression with Factorized Features

Image compression balances bit-rate and visual fidelity. Recent work recovers hidden details mainly via architectural tweaks—analysis transforms and entropy models [46, 51] or sophisticated decoders [88, 25]. We instead target the representation itself: our Factorized Features formulation (eq. (7)) explicitly models structural correlations, boosting quality.

**Priors from SR.** In addition, with our trained SR model described in section 4.2, it intuitively serves as a strong prior for information recovery, i.e., it contains extensive knowledge of how to reconstruct missing details and enhance image quality by leveraging learned patterns from the training data. Thus, since Super-Resolution and Image Compression share the core principle of reconstructing and enhancing image details from low-quality sources, we can effectively integrate this prior into the compression pipeline.

The overall pipeline is shown in fig. 5b. To demonstrate the robustness of our representation and the effectiveness of the SR prior, the compression and decompression networks greatly follow [62], with only the synthesis transform replaced by our SR pipeline, where the details can be referenced in the Supplementary Materials. In practice, the training is performed in two stages. After we obtain the trained SR prior, the model is fine-tuned with a lower learning rate alongside the compression module,

Table 1: **Quantitative comparisons on** $4\times$ **super-resolution with state-of-the-art methods.** The best results are colored **red**. The models with †are those who use the same-task pretraining [16], i.e., pretrained on ImageNet. Please refer to **quantitative results** in section 5.1 for details.

| Method | Params (M) | MACs (G) | Forward Pass Memory (MB) | Set5 PSNR↑ | Set5 SSIM↑ | Set14 PSNR↑ | Set14 SSIM↑ | B100 PSNR↑ | B100 SSIM↑ | Urban100 PSNR↑ | Urban100 SSIM↑ | Manga109 PSNR↑ | Manga109 SSIM↑ |
|---|---|---|---|---|---|---|---|---|---|---|---|---|---|
| EDSR [58](CVPR'17) | 43.09 | 207.07 | 1,182 | 32.46 | 0.8968 | 28.80 | 0.7876 | 27.71 | 0.7420 | 26.64 | 0.8033 | 31.02 | 0.9148 |
| RCAN [111](ECCV'18) | 15.59 | 65.52 | 1,176 | 32.63 | 0.9002 | 28.87 | 0.7889 | 27.77 | 0.7436 | 26.82 | 0.8087 | 31.22 | 0.9173 |
| SwinIR [55](ICCV'21) | 28.01 | 119.68 | 3,826 | 32.93 | 0.9043 | 29.15 | 0.7958 | 27.95 | 0.7484 | 27.56 | 0.8273 | 32.22 | 0.9273 |
| CAT-A+ [17](NIPS'22 Spotlight) | 16.60 | 70.29 | 3,508 | 33.14 | 0.9059 | 29.23 | 0.7968 | 28.01 | 0.7516 | 27.99 | 0.8356 | 32.52 | 0.9293 |
| ART [108](ICLR'23 Spotlight) | 16.56 | 69.94 | 3,010 | 33.04 | 0.9051 | 29.16 | 0.7958 | 27.97 | 0.7510 | 27.77 | 0.8321 | 32.31 | 0.9283 |
| ATD [110](CVPR'24) | 20.26 | 77.10 | 6,572 | 33.14 | 0.9061 | 29.25 | 0.7976 | 28.02 | 0.7524 | 28.22 | 0.8414 | 32.65 | 0.9308 |
| DAT [18](ICCV'23) | 14.80 | 61.66 | 4,192 | 33.15 | 0.9062 | 29.29 | 0.7983 | 28.03 | 0.7518 | 27.99 | 0.8365 | 32.67 | 0.9301 |
| RGT [19](ICLR'24) | 13.37 | 834.25 | 3,404 | 33.16 | 0.9066 | 29.28 | 0.7979 | 28.03 | 0.7520 | 28.09 | 0.8388 | 32.68 | 0.9303 |
| PFT [71](CVPR'25) | 19.66 | 80.21 | 6,414 | 33.15 | 0.9065 | 29.29 | 0.7978 | 28.02 | 0.7527 | 28.20 | 0.8412 | 32.63 | 0.9306 |
| HAT† [15](CVPR'23) | 20.77 | 86.02 | 3,692 | 33.18 | 0.9073 | 29.38 | 0.8001 | 28.05 | 0.7534 | 28.37 | 0.8447 | 32.87 | 0.9319 |
| ATD-L† | 49.42 | 184.83 | 15,582 | 33.15 | 0.9062 | 29.31 | 0.7985 | 28.02 | 0.7514 | 28.25 | 0.8422 | 32.78 | 0.9309 |
| ATD-F† (Ours) | 45.46 | 149.87 | 8,674 | 33.29 | 0.9082 | 29.48 | 0.8017 | 28.03 | 0.7539 | 28.53 | 0.8487 | 33.11 | 0.9335 |
| DAT-L† | 43.01 | 175.42 | 11,326 | 33.33 | 0.9084 | 29.40 | 0.8009 | 28.04 | 0.7543 | 28.49 | 0.8473 | 33.02 | 0.9321 |
| DAT-F† (Ours) | 40.00 | 134.42 | 6,206 | 33.45 | 0.9094 | 29.60 | 0.8039 | 28.13 | 0.7560 | 28.75 | 0.8520 | 33.23 | 0.9339 |
| HAT-L† | 40.84 | 167.27 | 6,804 | 33.30 | 0.9083 | 29.47 | 0.8015 | 28.09 | 0.7551 | 28.60 | 0.8498 | 33.09 | 0.9335 |
| HAT-F† (Ours) | 45.97 | 158.79 | 5,750 | 33.53 | 0.9100 | 29.65 | 0.8050 | 28.18 | 0.7569 | 28.79 | 0.8527 | 33.33 | 0.9342 |
| HAT-L-F† (Ours) | 66.04 | 240.03 | 8,888 | 33.75 | 0.9116 | 29.87 | 0.8091 | 28.31 | 0.7597 | 29.51 | 0.8637 | 33.36 | 0.9343 |
| HAT-F-Basis-First† (Ours, ablation) | 46.67 | 161.66 | 5,696 | 33.33 | 0.9085 | 29.47 | 0.8015 | 28.10 | 0.7554 | 28.57 | 0.8494 | 33.14 | 0.9336 |
| HAT-F-Concat† (Ours, ablation) | 45.52 | 129.05 | 4,826 | 33.46 | 0.9095 | 29.57 | 0.8035 | 28.16 | 0.7566 | 28.73 | 0.8518 | 33.28 | 0.9341 |

which is then trained end-to-end with common loss function used in learned image compression, defined as

$$L = R(\hat{y}) + R(\hat{z}) + \lambda \cdot D(x, \hat{x}), \tag{9}$$

where $\hat{z}$ is the hyperprior, $R(\cdot)$ refers to the bit-rate cost, $D(\cdot)$ is the distortion term, and $\lambda$ controls the trade-off.

#### 4.3.1 Multi-Image Compression

Intuitively, different images often share common frequency components. By learning a single, shared set of basis functions, we can leverage the frequently appearing similar or repetitive patterns across images to greatly reduce coding redundancy in multi-image compression. In practice, this means fewer total parameters and lower computational cost required when compressing an image collection.

Each per-image basis $b_i \in \mathbb{R}^{C_{b_i} \times H_{b_i} \times W_{b_i}}$ captures local pixel structure. We fuse the $M$ bases into a denoised, generic basis with a transformer $F_{\text{merge}}$ [83] applied at every position:

$$b_i(h, w) = F_{\text{merge}}\left(\{b_i^n(h, w) \mid n \in 1, ..., M\}\right), \tag{10}$$

Treating bases as tokens (prepended with a CLS token), the transformer's output forms the merged basis.

Because compression noise corrupts the coefficient map $X_{\text{coeff}}$ (fig. 5b), feeding it directly into the Basis Swin Transformer would amplify errors. Instead, we transmit the merged basis separately, alongside the quantized variables $Q(y, \mu), \hat{z}$, and reconstruct each image with this clean basis plus its own decoded coefficients, yielding higher quality at similar bit-rates. A single- vs. multi-image compression comparison is in the supplementary materials.

## 5 Experiments

In this section, we provide the experiment setup and performance of Super-Resolution and Image Compression tasks. Please refer to Supplementary Materials for more ablation studies, training, implementation details, and analysis.

### 5.1 Image Super-Resolution

**Experimental Setup.** Following the same-task pretraining strategy of [15, 21], we pretrain all Super-Resolution (SR) models on ImageNet[23]. Our Factorized Features plug a shared Basis Transformer into four pretrained SR backbones—SwinIR [55], HAT [16], DAT [18], and ATD [110]. Backbones

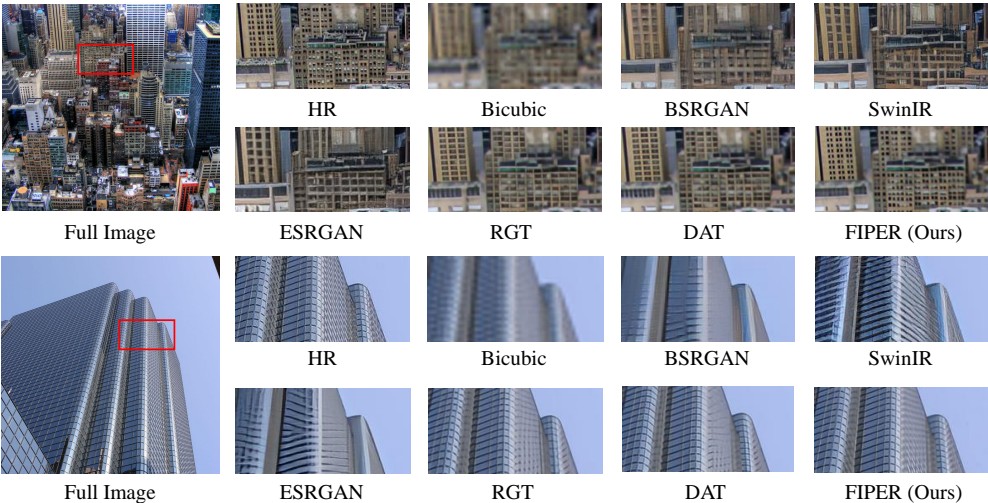

Figure 6: **Visual comparisons on super-resolution (4×).** The proposed method achieves the best reconstruction results compared to the reference HR images.

keep their original weights, while Basis Transformers start randomly. Pretraining runs 300k iterations on ImageNet with AdamW (lr 2e-4, batch 32, $\beta = 0.9/0.99$); finetuning runs 200k iterations on DF2K (DIV2K [1]+Flickr2K [59]) with lr 1e-5. Inputs are $256 \times 256$ crops, bicubically down-sampled to $64 \times 64$ for the backbone. We set the number of coefficient–basis pairs to $N = 6$ and use frequency scalars $\alpha_j \in \{1, 4, 16, 64\}$ to capture both low- and high-frequency details.

**Quantitative Results.** table 1 presents the quantitative comparison between our approach and state-of-the-art (SoTA) methods. We evaluate the methods using five benchmark datasets, including Set5 [6], Set14 [107], BSD100 [75], Urban100 [38], and Manga109 [76]. For quantitative metrics, PSNR and SSIM are reported. Our model delivers a **204.4%** average PSNR gain, defined as $(c-b)/(a-b)$ with $b$=SwinIR, $a$=HAT-L, and $c$=HAT-L-F. We embed three SoTA SR backbones: ATD [109], DAT [18], HAT [16], yielding ATD-F, DAT-F, and HAT-F. Each beats its parameter-matched baseline (ATD-L, DAT-L, HAT-L). As only HAT provides a large variant, we upscale ATD and DAT to the same size and train all models with identical pre-training and fine-tuning for fairness.

**The Order of Coefficient and Basis.** Furthermore, in traditional Fourier Series and other image processing methods [95], the basis is typically derived first and then used to compute the coefficients. In contrast, our method derives the coefficient features first, as illustrated in fig. 5a. To explore this difference, we develop another variant of our model, denoted **HAT-F-Basis-First**, where we reverse the order of operations. In this case, we first pass the image through the Basis Swin Transformer and then use the resulting basis features and the image input to derive the coefficients. This approach, however, leads to a gigantic performance drop, showing the importance of the order of the pipeline. Specifically, we argue that in our pipeline, the Coefficient Backbone functions more as a feature extraction module, where the refined features facilitate downstream basis extraction.

**The Importance of Factorized Features Reconstruction.** Lastly, to evaluate the effectiveness of our Factorized Features, we trained a model named **HAT-F-Concat**, which does not apply the formulation in eq. (7). Instead, it concatenates the basis and coefficient directly and decodes the resulting features to produce the output. Although this approach results in reduced performance, the Basis Swin Transformer with Sawtooth downsampling still contributes to improved reconstruction, even without Factorized Features Reconstruction, highlighting its effectiveness.

**Visual Comparison.** We provide the visual comparison in fig. 6. The images are randomly sampled from the DIV2K dataset. Our method faithfully reconstructs the image details, whereas the other approaches suffer from over-smoothing or hallucinating details absent in the ground truth.

## 5.2 Single- and Multi-Image Compressions

**Experimental Setup.** Image-compression experiments follow the SR protocol (section 5.1). Models are pretrained on ImageNet and fine-tuned for 200k steps on $256{\times}256$ crops. Compres-

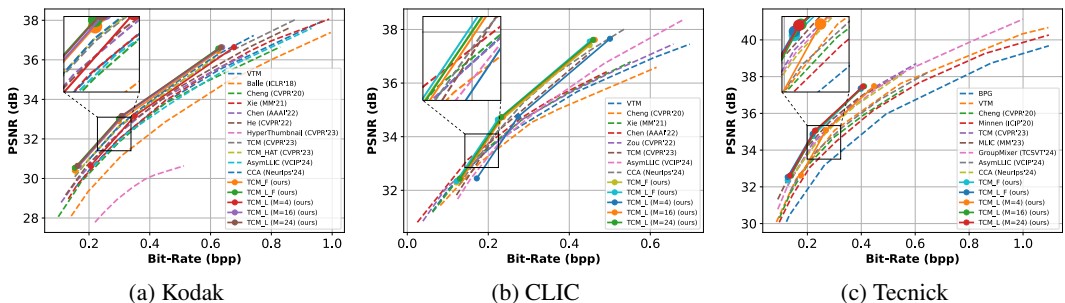

|  | (a) Kodak | (b) CLIC | (c) Tecnick |
|---|---|---|---|

Figure 7: **Performance (RD-Curve) evaluation on image compression using different datasets.**

Table 2: **Comprehensive evaluation for image compression.** Using VTM as an anchor for calculating BD-Rate. Latencies are measured under an NVIDIA GTX 3090 GPU.

| Method | BD-Rate (%) ↓ | Total Encoding Time (s) ↓ | Total Decoding Time (s) ↓ | Params(M) |
|---|---|---|---|---|
| VTM | 0.00 | 129.21 | 0.14 | - |
| Cheng (CVPR'20)[20] | 5.44 | 1.98 | 4.69 | 29.6 |
| Xie (MM'21)[101] | -0.78 | 2.93 | 6.00 | 50.0 |
| STF (CVPR'22)[116] | -4.31 | 0.14 | 0.13 | 99.9 |
| ELIC (CVPR'22)[36] | -7.24 | 0.07 | 0.09 | 36.9 |
| TCM (CVPR'23)[62] | -11.74 | 0.16 | 0.15 | 76.7 |
| SegPIC (ECCV'24) [67] | -8.26 | 0.14 | 0.13 | 83.5 |
| LALIC (CVPR'25) [27] | -15.26 | 0.27 | 0.15 | 63.2 |
| TCM-HAT-L-F (Ours) | -21.09 | 0.109 | 0.264 | 110.34 |
| TCM-HAT-F-multi M=1 (Ours) | 27.96 | 0.2320 | 0.1742 | 131.35 |
| TCM-HAT-F-multi M=2 (Ours) | 2.70 | 0.1998 | 0.1270 | 131.35 |
| TCM-HAT-F-multi M=4 (Ours) | -10.11 | 0.1846 | 0.1039 | 131.35 |
| TCM-HAT-F-multi M=8 (Ours) | -16.61 | 0.1756 | 0.0922 | 131.35 |
| TCM-HAT-F-multi M=16 (Ours) | -19.88 | 0.1715 | 0.0863 | 131.35 |
| TCM-HAT-F-multi M=24 (Ours) | -20.97 | 0.1702 | 0.0844 | 131.35 |

sion/decompression networks start from TCM weights [62] and are optimized end-to-end with AdamW (lr 1e-5, batch 16, $\beta$ = 0.9/0.99) [72]. Integrating our SR module into TCM yields TCM-HAT-F and TCM-HAT-L-F; TCM-HAT-F-multi is the multi-image variant.

**Rate-Distortion Performance Comparison.** We compare our model with State-of-the-Art learned end-to-end image compression algorithms, including [63], [13], [116], [101], [20], [3], [51], [40], [79], [5], [84], and [36]. The classical image compression codec, VVC [93], is also tested by using VTM12.1.The rate-distortion performance on various datasets, including Kodak, Tecnick's old test set with resolution $1200\times1200$, and CLIC Professional Validation, is shown in fig. 7. TCM-HAT-L-F achieves a -21.09% BD-Rate vs. VTM (Table 2), surpassing earlier work. Multi-image mode (TCM-HAT-F-multi) scales well, reaching -20.97% at $M$=24: sending a shared basis curbs Coefficient-to-Basis error, though distortion grows slightly as basis capacity is stretched. Across Kodak, CLIC, and Tecnick, FIPER consistently boosts rate-distortion and maintains competitive latency.

## 6 Conclusion

We proposed Factorized Features, a representation that decomposes images into multi-frequency components to model implicit structures and patterns. Our approach addresses challenges in Super-Resolution and Image Compression by effectively restoring details and preserving visual fidelity. We integrate SR priors with Image Compression for better information recovery and introduce a basis-merging technique to enhance multi-image processing performance.

**Limitations.** Although effective, our method requires further optimization for computation-limited scenarios such as real-time decoding. Additionally, incorporating semantic information remains a promising direction for future research.

**Acknowledgements.** This research was funded by the National Science and Technology Council, Taiwan, under Grants NSTC 112-2222-E-A49-004-MY2, 113-2628-E-A49-023-, Academia Sinica under Grant AS- CDA-110-M09, and Yin Shu-Tien Foundation Taipei Veterans General Hospital-National Yang Ming Chiao Tung University Excellent Physician Scientists Cultivation Program, 114-Y-B-144. The authors are grateful to Google, NVIDIA, and MediaTek Inc. for their generous donations, and we would also like to express our sincere gratitude to Jia-Wei Liao for valuable assistance in paper writing and revision. Yu-Lun Liu acknowledges the Yushan Young Fellow Program by the MOE in Taiwan.

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

# A  Overview

This supplementary material provides additional implementation details and experimental results to complement our main manuscript. Our FIPER framework explicity models multi-scale patterns and captures periodic features effectively, as visualized in fig. 8.

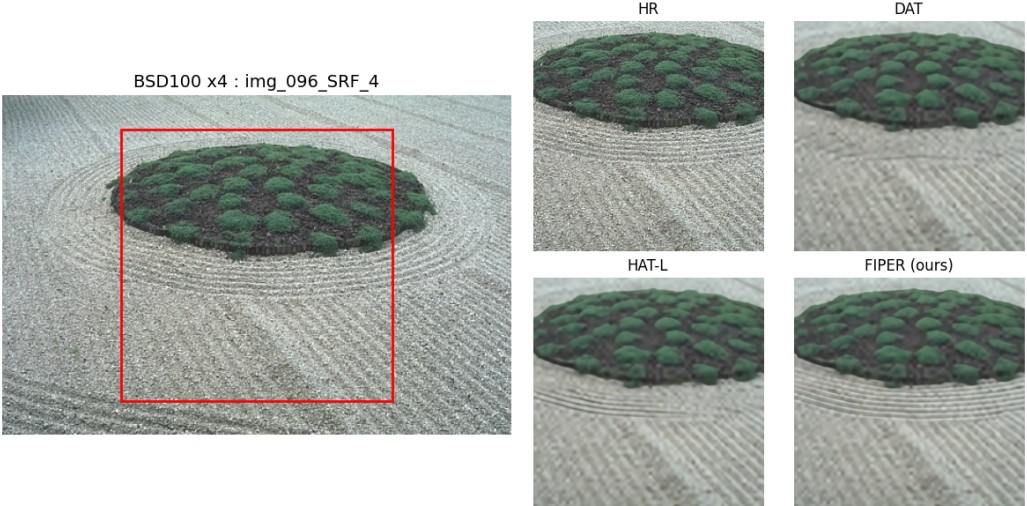

Figure 8: **Visual comparisons on super-resolution (4×).**

In the following, we dive into the details. Specifically, we first elaborate on coordinate transformation, the background of learned image compression, and a comparison between single- and multi-image compression in appendix B, which were omitted in the main paper due to space limitations. Following this, we discuss the differences with factor fields [12] in appendix C. Next, in appendix D, we present comprehensive details about the experiments and benchmarks, including the Video Super-Resolution task, which was not discussed in the main paper. Then, we provide ablation studies regarding different components of our pipeline, validating the design in appendix E. Finally, we provide extensive visualizations to facilitate a more detailed comparison of the results in appendix H.

# B  Method Details

In this section, we provide the details of Sawtooth coordinate transformation and downsampling inspired by it, the background of Learned Image Compression, and comparison between single- and multi-image compression.

### B.1  Sawtooth Coordinate Transformation and Downsampling

**Sawtooth transformation**    Sawtooth transformation is formulated as $\gamma(x) = x \mod k, \ k \in R$. We can easily observe that such transformation implicitly captures patch-like frequency information as shown in fig. 9a, and thus, we propose that by leveraging the inter-patch information from the sawtooth coordinate transformation, the visual correspondence between spatial locations can be effectively represented.

**Downsampling based on Sawtooth transformation**    To reconstruct a $256 \times 256$ image, we formulate our Factorized Features as

$$\hat{I}(x) = \mathcal{P}\Big(\mathbf{Concat}_{i=1}^{N}\,_{j=1}^{K}\Big\{c_{ij}(x) \odot \psi(\alpha_j \cdot b_i(\gamma_i(x)))\Big\}\Big), \tag{11}$$

which is the same as eq. (7) in the main paper.

In practice, the bases are of shape $C \times H \times W$ with $N = 6$. We set $H = W = \{256, 128, 64, 32, 16, 8\}$ and $C = 24$ for all bases. Thus, we sample each basis with $\gamma_i(x) = x$

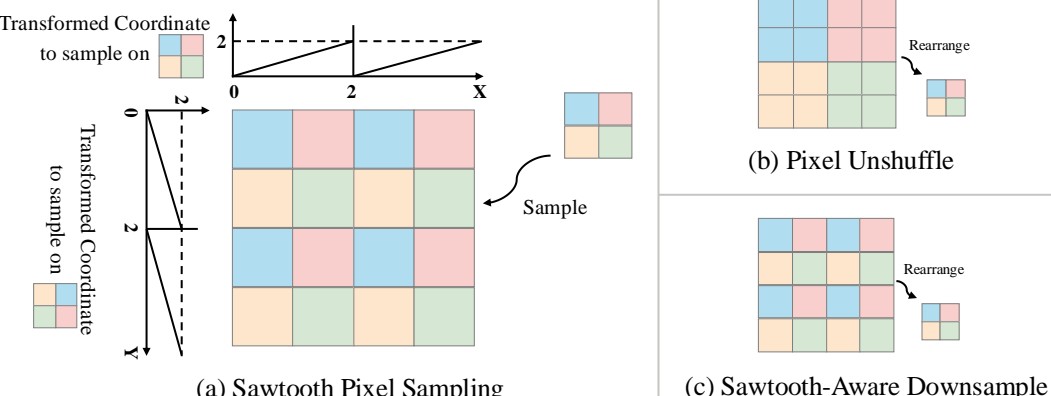

Figure 9: **The correlation between coordinate transformation and downsampling.** (a) The sawtooth transformation example with $k = 2$. (b) The PixelUnShuffle downsample. (c) To explicitly model the information for sampling with a sawtooth, we rearrange the feature map in a dilation-like manner in the downsample layer of the Basis Swin Transformer. This way, the feature sampled would capture the information in the original layout correctly; in other words, $I = \text{Sawtooth-Pixel-Sampling}(\text{Sawtooth-Aware-Downsample}(I))$

mod $k_i$ where $x \in \{0, ..., 255\}^2$ and $k = \{256, 128, 64, 32, 16, 8\}$. This formulation can be interpreted as we sample the same feature on different spatial location of a image, e.g., when using $b_1 \in R^{24 \times 128 \times 128}$ with $k_1 = 128$, $x = \{0, 0\}$ and $x = \{128, 0\}$ will be extracting $b_1(0, 0)$, which is also visualized in fig. 9a. Therefore, we want to aggregate the information appearing in different location of the image, where these spatial locations will be sampled at the same location on the basis; that is, we want the information of $I(0, 0)$ and $I(128, 0)$ to be aggregated in $b_1(0, 0)$ in previous example. This way, we can explicitly model repetitve structure in different scale, and such frequency modeling subsequently enhances the reconstruction.

Based on the analysis, as visualized in fig. 9c, we integrate **Sawtooth-Aware Downsample** layer into Basis Transformer. Such operation is essentially the inverse of Sawtooth transformation, i.e., we merge the feature together with Sawtooth-Aware Downsample, decode the feature to basis, and then sample the basis to different spatial location via Sawtooth transformation.

### B.2 Learned Image Compression.

Following [79, 3], a learned image compression model with a channel-wise entropy model can be formulated as:

$$
\begin{aligned}
z = h_a(y; \phi_h), \; y = g_a(x; \phi), \\
\{F_{\text{mean}}, F_{\text{scale}}\} = h_s(\hat{z}; \theta_h), \; \hat{z} = Q(z), \\
\hat{y} = \{Q(y_0 - \mu_0) + \mu_0, ..., Q(y_t - \mu_t) + \mu_t\}, \\
\hat{x} = g_s(\overline{y}; \theta), \; \overline{y} = \textbf{Refine}_{\theta_r}(\mu_0, ..., \mu_t, \hat{y}),
\end{aligned}
\tag{12}
$$

where $0 <= t < l$, $\mu_t = e_i(\hat{y}_{<i}, F_{\text{mean}})$. The encoder $g_a$ transforms the raw image $x$ into a latent representation $y$. A hyper-prior encoder $h_a$ further processes $y$ to output $z$, capturing spatial dependencies. $z$ is then quantized to $\hat{z}$, where a factorized density model $\varphi$ is used to encode quantized $\hat{z}$ as $p_{\hat{z}|\varphi}(\hat{z} \mid \varphi) = \prod_j \left( p_{z_j|\varphi}(\varphi) * \mathcal{U}\left(-\frac{1}{2}, \frac{1}{2}\right) \right)(\hat{z}_j)$ where $j$ specifies the position or each element of each signal. The derivation of the equation can be found in [2]. Next, $\hat{z}$ is decoded by $h_s$ to produce features $F_{\text{mean}}$ and $F_{\text{scale}}$, used to estimate the mean $\mu$ and variance $\sigma$ of $y$. The latent $y$ is divided into $l$ slices, and each quantized around computed means $\mu_t$. These $\mu_t$ are derived from earlier quantized slices and $F_{\text{mean}}$ by a slice network $e_i$. The quantized slices form $\hat{y}$. For decompression, $\hat{y}$ is refined using $\textbf{Refine}_{\theta_r}$ based on $\mu_t$ and $\hat{y}$ to produce $\overline{y}$, approximating the original $y$. Finally, $g_s$ reconstructs the decompressed image $\hat{x}$ from $\overline{y}$. The model is trained using a

Lagrangian multiplier-based rate-distortion optimization:

$$
\begin{aligned}
L &= R(\hat{y}) + R(\hat{z}) + \lambda \cdot D(x, \hat{x}) \\
&= \mathbb{E}\left[-\log_2\left(p_{\hat{y}|\hat{z}}\left(\hat{y} \mid \hat{z}\right)\right)\right] + \mathbb{E}\left[-\log_2\left(p_{\hat{z}|\varphi}\left(\hat{z} \mid \varphi\right)\right)\right] \\
&\quad + \lambda \cdot \mathcal{D}\left(x, \hat{x}\right)
\end{aligned}
\tag{13}
$$

where $R(\hat{y})$ and $R(\hat{z})$ denote bit rates, $D(x, \hat{x})$ is the distortion term (calculated by MSE), and $\lambda$ balances compression efficiency and image fidelity. In our experiments, we follow [62], modifying only $g_s$ to demonstrate our representation's effectiveness.

## B.3 Comparison between single- and multi-image compression

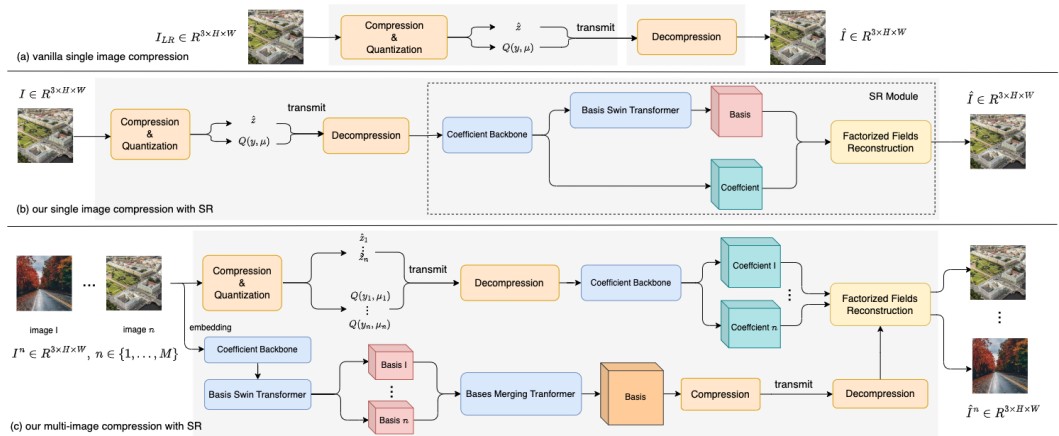

Figure 10: **The illustration of our image-compression framework (a)** Traditional learning-based compression methods. **(b)** Our approach surpasses (a) by incorporating our Factorized Features and Super-Resolution (SR) Module from section 4.2 in the main paper as information-recovery prior. **(c)** Expanding on (b), we introduces a multi-image compression strategy that utilizes both our SR Module and a Basis Merging Transformer to capture shared structure.

We provide visual comparison of single- and multi-image compression in fig. 10b and fig. 10c, respectively, with traditional compression method in fig. 10a. First, we have single image compression(SIC), the same as fig. 5b in the main paper. The goal of the SIC pipeline is to validate two aspects: (1) that Factorized Features, which are designed to represent images, can be applied to different tasks; (2) that leveraging super-resolution (SR) priors improves reconstruction quality. Note that we directly adopt the SR module's architecture to also reduce overall complexity.

Next, we derive multi-image compression (MIC) as in section 4.3.1 of the main paper. In traditional signal processing, a shared set of bases is often used to represent diverse sources. We adopt this property to further reduce compression redundancy. Specifically, our approach transmits (1) image features, which are used to decode coefficients, and (2) the bases themselves. By contrast, single image compression in fig. 10b transmits only image features. Intuitively, transmitting the bases provides additional information that can improve reconstruction quality. However, directly transmitting all bases would significantly increase the bit rate. To address this, we merge the bases into a single set before transmission, thereby enhancing reconstruction while minimizing extra transmission costs.

## C Differences from Factor Fields

We **explicitly highlight** how **Sawtooth-based sampling** and **Multi-frequency Modulation** extend the original factor fields. In 2D toy examples as shown in fig. 11, traditional SR/IC representations can be viewed as a "vanilla" scenario with $N = 1$ and $c(x) = 1$. factor fields already improve accuracy over such baselines, but our introduction of $\psi$ and $\alpha$ further captures both high- and low-frequency details. Specifically, forcing $\alpha = 4$ compels the network to learn broader frequency ranges and avoid noise from the fixed modulation. To confirm these advantages, we trained *smaller SR models* and compared them with factor fields in table 3. The results illustrate that our multi-frequency constraint

yields significant gains even in simpler configurations, underscoring its importance for high-frequency detail preservation and overall reconstruction quality.

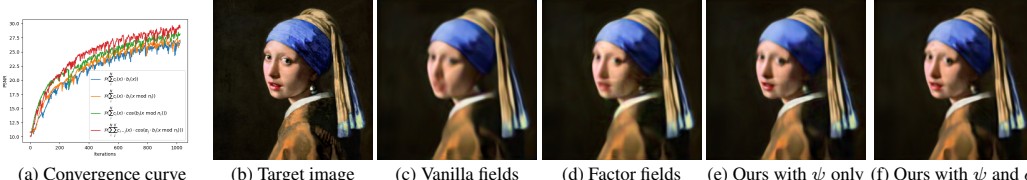

(a) Convergence curve     (b) Target image     (c) Vanilla fields     (d) Factor fields     (e) Ours with $\psi$ only   (f) Ours with $\psi$ and $\alpha$

Figure 11: **2D toy example.**

Table 3: **Comparison with Factor Fields.** HAT-FA-Small denotes replace our Factorized Features with factor fields.

| Method | Sawtooth $\gamma(x)$ | $\psi$ | $\alpha$ | Params (M) | MACs (G) | Forward Pass Memory (MB) | Set5 PSNR↑ | Set5 SSIM↑ | Set14 PSNR↑ | Set14 SSIM↑ | B100 PSNR↑ | B100 SSIM↑ | Urban100 PSNR↑ | Urban100 SSIM↑ | Manga109 PSNR↑ | Manga109 SSIM↑ |
|---|---|---|---|---|---|---|---|---|---|---|---|---|---|---|---|---|
| HAT-S | | | | 9.62 | 40.38 | 3,214 | 32.92 | 0.9047 | 29.15 | 0.7958 | 27.97 | 0.7505 | 27.87 | 0.8346 | 32.35 | 0.9283 |
| HAT | | | | 20.77 | 86.02 | 3,692 | 33.04 | 0.9056 | 29.23 | 0.7973 | 28.00 | 0.7517 | 27.97 | 0.8368 | 32.48 | 0.9292 |
| HAT-FA-Small | ✓ | | | 15.99 | 57.61 | 3,678 | 33.06 | 0.9065 | 29.26 | 0.7976 | 28.04 | 0.7524 | 28.12 | 0.8413 | 32.52 | 0.9301 |
| HAT-F-Small-only-sin | ✓ | {sin} | {1, 4, 16, 64} | 16.13 | 66.34 | 4,176 | 33.16 | 0.9067 | 29.33 | 0.7989 | 28.05 | 0.7530 | 28.26 | 0.8429 | 32.64 | 0.9312 |
| HAT-F-Small-half-freq | ✓ | {sin, cos} | {1, 16} | 16.13 | 66.34 | 4,148 | 33.18 | 0.9073 | 29.35 | 0.8003 | 28.02 | 0.7521 | 28.31 | 0.8431 | 32.70 | 0.9316 |
| HAT-F-Small | ✓ | {sin, cos} | {1, 4, 16, 64} | 16.26 | 75.06 | 4,640 | 33.24 | 0.9079 | 29.39 | 0.8009 | 28.06 | 0.7543 | 28.41 | 0.8453 | 32.82 | 0.9324 |
| HAT-F (Our FIPER) | ✓ | {sin, cos} | {1, 4, 16, 64} | 45.97 | 158.79 | 5,750 | 33.31 | 0.9085 | 29.45 | 0.8011 | 28.09 | 0.7553 | 28.55 | 0.8463 | 32.99 | 0.9330 |
| HAT-L (ImageNet pre-trained) | | | | 40.84 | 167.27 | 6,804 | 33.30 | 0.9083 | 29.47 | 0.8015 | 28.09 | 0.7551 | 28.60 | 0.8498 | 33.09 | 0.9335 |

# D    Implementation Details

**Single Image Super-Resolution**    The architecture of our SR networks is shown in fig. 12. Firstly, for Coefficient Backbones from various pre-trained models like [15, 109, 18], we replace the output upsampling layer (mostly pixel shuffle) with convolution of channel $D = 256$, i.e., for input image with resolution $64 \times 64$, the original pre-trained models process hidden states in $64 \times 64$, and then upsample the output to $256 \times 256$ for final result, where we replace the upsampling with convolution and output the final hidden state $X_{coeff} \in R^{256 \times 64 \times 64}$.

Next, the Basis Swin Transformer begins with feature embedding, extending the channels $D$ from 256 to 384, where the features are then passed through a series of Swinv2[70] and Sawtooth-Aware Downsampling Blocks. Specifically, each block contains two layers, one of which uses shifted window attention. After each block, the hidden state goes through upsampling by 4x and convolution to convert the output basis channel to 24. Then, the Sawtooth-Aware Downsample block rearranges the hidden state $h$ as discussed in the main paper, which can be viewed as:

$$h = \textbf{rearrange}(h, \text{'b c (s h) (s w)} \rightarrow \text{b (s s c) h w'}), \ s = 2 \qquad (14)$$

, and reduces the feature channel by 4. In all experiments we use $N = 6$, i.e. there are 6 bases, where $Basis_i \in R^{24 \times \frac{256}{2^i} \times \frac{256}{2^i}}$ for i in $\{0 \dots 5\}$.

Finally, we derive our Coefficient by a series of Convolution and Pixelshuffle layers and decoded the Bases and Coefficient to a predicted high-resolution image by eq. (15):

$$\hat{I}(x) = \mathcal{P}\Big(\textbf{Concat}_{i=1}^{N} \ _{j=1}^{K} \Big\{ c_i(x) \odot \psi(\alpha_j \cdot b_i(\gamma_i(x))) \Big\}\Big). \qquad (15)$$

In practice, the coefficient is generated with channels equal to the total number of base channels, and it is split accordingly by channels.

**Multi-Image (Video) Super-Resolution**    Building upon our prior work in Single-Image Super-Resolution (SISR), we extend the applicability of Factorized Features to tackle the challenges in Multi-Image Super-Resolution (MISR). In the main paper, we introduced the Basis Merging Transformer for Multi-Image Compression. Here, we further expand its functionality to generate per-timestamp Coefficients and Bases.

To evaluate the effectiveness of our approach, we use Video Super-Resolution (VSR) as the benchmark task. Specifically, we simply leverage transformer blocks from [83] to model temporal correlations.

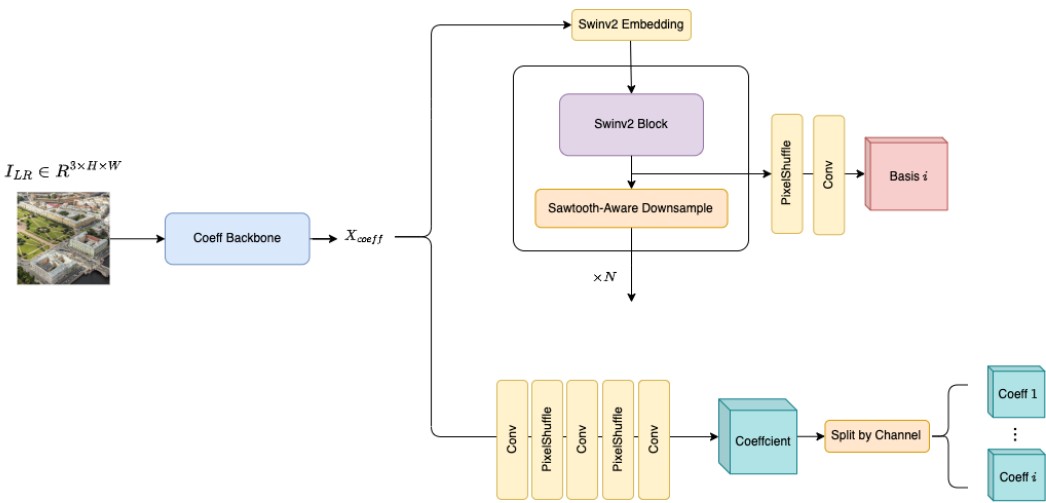

Figure 12: **Detailed architecture of Super-Resolution Modules**

| Method | Frames REDS/Vimeo | REDS4 PSNR | REDS4 SSIM | Vimeo-90K-T PSNR | Vimeo-90K-T SSIM | Vid4 PSNR | Vid4 SSIM |
|---|---|---|---|---|---|---|---|
| BasicVSR++[10] | 30/14 | 32.39 | 0.9069 | 37.79 | 0.9500 | 27.79 | 0.8400 |
| VRT[57] | 16/7 | 32.19 | 0.9006 | 38.20 | 0.9530 | 27.93 | 0.8425 |
| RVRT[56] | 30/14 | 32.75 | 0.9113 | 38.12 | 0.9527 | 27.99 | 0.8462 |
| PSRT[87] | 16/14 | 32.72 | 0.9106 | 38.27 | 0.9536 | 28.07 | 0.8485 |
| IA-RT[102] | 16/7 | 32.90 | 0.9138 | 38.14 | 0.9528 | 28.26 | 0.8517 |
| Ours-HAT-F | 16/7 | 32.67 | 0.9096 | 38.09 | 0.9512 | 27.95 | 0.8441 |

Table 4: Performance comparison among different methods on REDS4, Vimeo-90K-T, and Vid4 datasets. **Note that we don't use any special design on the network, such as temporal correspondence or motion sampling like previous works, demonstrating our effectiveness.**

For a given video, we first derive the Coefficient and Basis for each frame and rearrange each level $i$ of the Coefficients or Basis such that the spatial dimensions $H$ and $W$ are treated as the batch dimension, while the temporal dimension is processed as a sequence of tokens for the Transformer. Next, we apply positional embedding to the tokens, process the tokens with the transformer blocks, and finally decode the Coefficient and Basis token from the respective position with convolution layers for the output super-resolved frames.

table 4 shows quantitative comparison with state-of-the-art methods. All VSR experiments were conducted using bicubic 4X downsampling. The training dataset includes the REDS[82] and Vimeo-90K[103] datasets, while the testing dataset comprices REDS4[82], Vid4[60], and Vimeo-90K-T[103].

For the REDS dataset, we train for 300k iterations using 16 input frames, with a learning rate of 1e-4 and a cosine learning rate that gradually decays to 1e-7. The batch size is set to 8, and the Coefficient Backbone and the Basis Swin Transformer are initialized from SISR. When training on the Vimeo-90K dataset, we first conduct 300k iterations with 14 input frames with flip sequence, we then train model on 7 input frames with flip sequence, using a learning rate of 1e-4 and a cosine learning rate decay to 1e-7. We initialize the weights using the model trained on the REDS dataset. The batch size remains 8. Note that the training is only conducted on the newly added transformer blocks, i.e., the Coefficient Backbone and Basis Swin Transformer are frozen. Test results for the REDS model are reported on the REDS4 dataset, while test results for the Vimeo-90K model are reported on Vimeo-90K-T and Vid4. We calculate PSNR and SSIM on the RGB channel for REDS4 and Y channel for Vimeo-90K-T and Vid4, following previous work[102, 87, 56].

**Single Image Compression**    As discussed in the main paper, to make a fair comparison with the state-of-the-art, our entropy model and compression backbone greatly follow that of [62], as shown

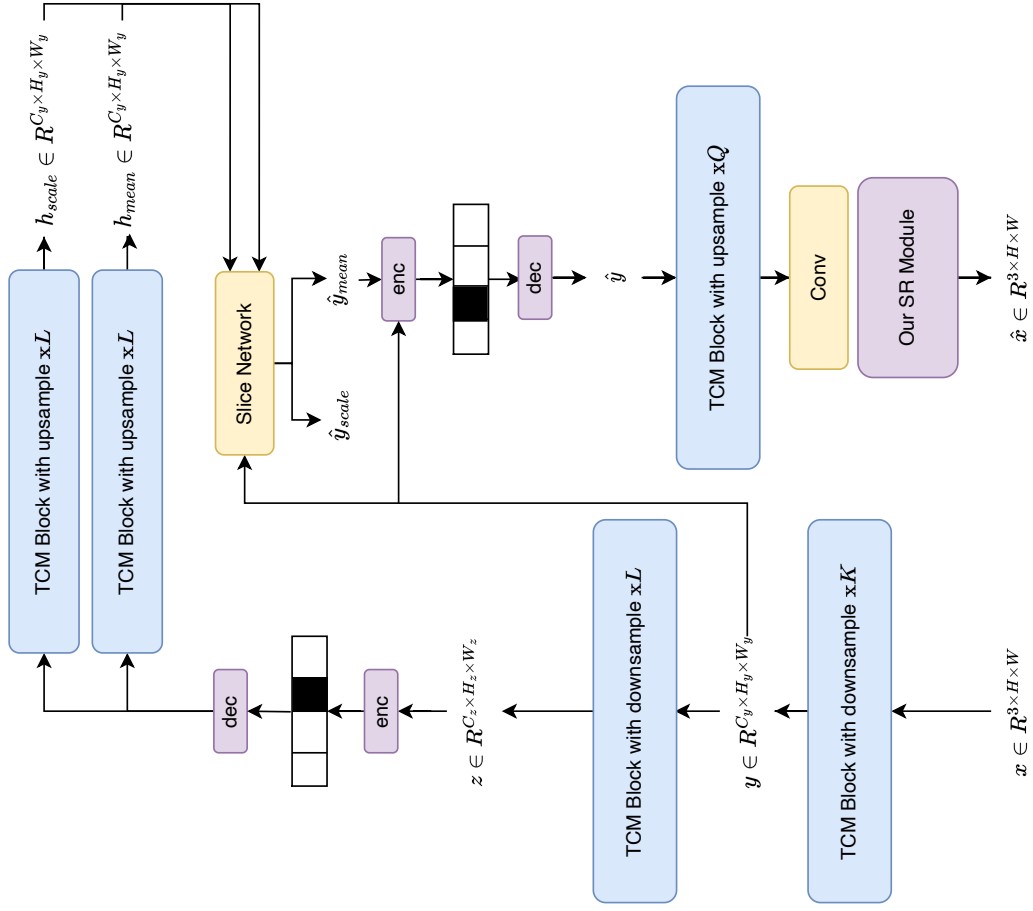

Figure 13: **Detailed architecture of Compression Modules**

in fig. 13(more details can be bound in [62]. What we do here is that we use the output from the last main layer and convert it to match the hidden channel of the SR Module.

**Multi-Image Compression**    Following VSR, we use transformer blocks to merge the bases. The difference here is that we additionally prepend a CLS token so that we can use it as the final merged basis, i.e., if we have 16 bases of $W = H = 64$ to merge into one, we rearrange the $64 \times 64$ tokens as batch dimension, use the $16 + 1$ as sequence dim, and finally treat the additional 1(CLS) token as output. For the merged basis compression and transmission, we utilize the same architecture of the image branch, i.e., we randomly initialize a module from [62] with input and out channels change. After the Basis Merging Transformer is trained, we follow the same training setting for Basis compression while randomly sampling 1 to 24 images for a batch and setting only the Basis compression module trainable.

## E    Ablation Studies

### E.1    Merged-Basis Strategy & SR Priors.

The SR prior accelerates detail recovery by leveraging high-resolution knowledge while basis merging exploits shared structures for efficiency. Since merging is introduced *after*, the compression model is pre-trained, and improvements stem not only from the SR prior but also from the well-initialized basis merging. When trained from scratch (see table 5), the merging transformer tends to find a universal solution rather than image-specific structures. Additionally, merging inevitably causes information loss, prompting the coefficient branch to encode more image-specific details. This mutual offset between SR prior and basis merging ultimately strengthens our approach.

| | TCM-HAT-F-multi $M=24$ | TCM-HAT-F-multi $M=24$ Coeff-Frozen | TCM-HAT-F-multi $M=24$ Scratch |
|---|---|---|---|
| BD-rate % | -20.97 | -19.18 | -18.34 |

Table 5: **Comparison of Merged-Basis Strategies.**

## E.2 Encoding/Decoding Times.

Fig. 5(c) processes each of the $M$ images through the Coefficient Backbone, Basis Swin Transformer, and Basis Merging Transformer, then compresses and transmits the merged basis. In Tab. 2, different $M$ values primarily affect the run time of the Basis Merging Transformer. Although merging larger $M$ increases computation, *per-image* overhead remains small once averaged by $M$, as shown in table 6. Notably, Tab. 2 assumes each image's basis is compressed independently, which can be confusing. In practice, a merged basis is processed only *once*, so averaged encoding/decoding time is more realistic.

| | $M=1$ | $M=2$ | $M=4$ | $M=8$ | $M=16$ | $M=24$ |
|---|---|---|---|---|---|---|
| Total Basis Merging Time (sec) | 0.00328 | 0.00628 | 0.01140 | 0.02312 | 0.04784 | 0.08064 |
| Averaged Basis Merging Time (sec) by $M$ | 0.00328 | 0.00314 | 0.00285 | 0.00289 | 0.00299 | 0.00336 |
| Averaged Merged Basis Compression Time (sec) by $M$ | 0.0645 | 0.0323 | 0.0161 | 0.0081 | 0.0040 | 0.0027 |
| Averaged Merged Basis Decompression Time (sec) by $M$ | 0.0935 | 0.0467 | 0.0234 | 0.0117 | 0.0058 | 0.0039 |
| Adjusted Enc Time (sec) by $M$ | 0.2320 | 0.1998 | 0.1836 | 0.1756 | 0.1715 | 0.1702 |
| Adjusted Dec Time (sec) by $M$ | 0.1740 | 0.1272 | 0.1039 | 0.0922 | 0.0863 | 0.0844 |
| BD-Rate % | 27.96 | 2.70 | -10.11 | -16.61 | -19.88 | -20.97 |

Table 6: **Comparison of Encoding/Decoding Times.**

## E.3 Inference Time Comparison for SR.

We provide the inference time in table 7, compared with extensive SR models. The inference time of different components of FIPER is also provided.

| | ATD | DAT | RGT | HAT-S | HAT | HAT-L | HAT-F(Ours) | Coefficient Backbone (HAT-F) | Basis Swin Transformer (HAT-F) | Factorized Features Reconstruction (HAT-F) |
|---|---|---|---|---|---|---|---|---|---|---|
| Inference Time (s) | 0.0592 | 0.0594 | 0.0612 | 0.0387 | 0.0398 | 0.0778 | 0.0716 | 0.0400 | 0.0132 | 0.0184 |
| PSNR ont Set5 | 33.14 | 33.15 | 33.16 | 32.92 | 33.18 | 33.30 | 33.53 | - | - | - |

Table 7: **Inference time comparison for SR.**

## E.4 Influence of SR Priors in Image Compression.

We conduct experiments with various configurations to verify our image compression pipeline's effectiveness. TCM-HAT refers to using the original HAT [16] instead of our SR Module in fig. 10b. TCM-HAT-F represents our full pipeline, while TCM-HAT-F-Scratch initializes the SR Module randomly. Results demonstrate that integrating SR priors improves image compression performance, and our representation further enhances results, highlighting the robustness of Factorized Features.

Table 8: **Validation of the effectiveness of SR prior.** The best PSNRs are marked in red.

| Method | Kodak | | | | | | CLIC | | | | | | Tecnick | | | | | |
|---|---|---|---|---|---|---|---|---|---|---|---|---|---|---|---|---|---|---|
| | $\lambda=0.0025$ | | $\lambda=0.0067$ | | $\lambda=0.025$ | | $\lambda=0.0025$ | | $\lambda=0.0067$ | | $\lambda=0.025$ | | $\lambda=0.0025$ | | $\lambda=0.0067$ | | $\lambda=0.025$ | |
| | bpp | PSNR↑ | bpp | PSNR↑ | bpp | PSNR↑ | bpp | PSNR↑ | bpp | PSNR↑ | bpp | PSNR↑ | bpp | PSNR↑ | bpp | PSNR↑ | bpp | PSNR↑ |
| TCM[62] | 0.1533 | 30.0834 | 0.2983 | 32.5841 | 0.6253 | 36.1345 | 0.1214 | 31.8207 | 0.2235 | 34.2098 | 0.4503 | 37.1201 | 0.1268 | 32.0588 | 0.2193 | 34.3669 | 0.3981 | 36.9066 |
| TCM-HAT-F-Scratch | 0.1570 | 30.0857 | 0.2976 | 32.5893 | 0.6211 | 36.1389 | 0.1214 | 31.9421 | 0.2235 | 34.2894 | 0.4503 | 37.1434 | 0.1258 | 32.0632 | 0.2189 | 34.3781 | 0.4001 | 36.9223 |
| TCM-HAT | 0.1567 | 30.1843 | 0.2992 | 32.6454 | 0.6268 | 36.2267 | 0.1220 | 31.9737 | 0.2266 | 34.3319 | 0.4512 | 37.2486 | 0.1262 | 32.1423 | 0.2174 | 34.5124 | 0.3971 | 36.9934 |
| TCM-HAT-F | 0.1574 | 30.4012 | 0.2998 | 32.8910 | 0.6276 | 36.4461 | 0.1229 | 32.1917 | 0.2249 | 34.4109 | 0.4512 | 37.3135 | 0.1255 | 32.4591 | 0.2186 | 34.7656 | 0.3975 | 37.3244 |

## E.5 Effectiveness of Factorized Features Design.

We conduct experiments to verify our multi-frequency modulation. The quantitative performance reported on single-image regression is shown in table 9, where each result is measured after 256 iterations. Compared to baseline results, our refinements in modeling pixel-level frequency have significantly improved all performance metrics. Additionally, our results demonstrate that the modulation function $\psi$ and the scalar $\alpha$ are interdependent, each essential to the other's function.

Table 9: **Comparison of improvements of Factorized Features.** $\psi$ and $\alpha$ are the same in eq. (8). Note that here we test on single-image regression

| Metric | PSNR ↑ | SSIM ↑ | LPIPS ↓ |
|---|---|---|---|
| Baseline [12] | 22.04 | 0.505 | 0.5296 |
| Ours | 38.44 | 0.999 | 0.0385 |
| No $\psi$ to control magnitude | 13.46 | 0.147 | 0.766 |
| No $\alpha$ for pixel-wise frequency information | 21.25 | 0.537 | 0.527 |

| Method | Latency (ms) | #FLOPs (G) | #Params (K) | Set5 PSNR / SSIM | Set14 PSNR / SSIM | B100 PSNR / SSIM | Urban100 PSNR / SSIM | Manga109 PSNR / SSIM |
|---|---|---|---|---|---|---|---|---|
| SwinIR-lt | 222.9 | 63.6 | 930 | 32.44 / 0.8976 | 28.77 / 0.7858 | 27.69 / 0.7406 | 26.47 / 0.7980 | 30.92 / 0.9151 |
| ELAN-lt | 18.0 | 54.1 | 640 | 32.43 / 0.8975 | 28.78 / 0.7858 | 27.69 / 0.7406 | 26.54 / 0.7982 | 30.92 / 0.9150 |
| OmniSR | 22.5 | 50.9 | 792 | 32.49 / 0.8988 | 28.78 / 0.7859 | 27.71 / 0.7415 | 26.64 / 0.8018 | 31.02 / 0.9150 |
| SRFormer-lt | 287.2 | 62.8 | 873 | 32.51 / 0.8988 | 28.82 / 0.7872 | 27.73 / 0.7422 | 26.67 / 0.8032 | 31.17 / 0.9165 |
| ATD-lt | 189.7 | 100.1 | 769 | 32.63 / 0.8998 | 28.89 / 0.7886 | 27.79 / 0.7440 | 26.97 / 0.8107 | 31.48 / 0.9198 |
| HiT-SRF | 82.1 | 58.0 | 866 | 32.55 / 0.8999 | 28.87 / 0.7880 | 27.75 / 0.7432 | 26.80 / 0.8069 | 31.26 / 0.9171 |
| ASID-D8 | 61.8 | 49.6† | 748 | 32.57 / 0.8990 | 28.89 / 0.7898 | 27.78 / 0.7449 | 26.89 / 0.8096 | – |
| MambaIR-lt | 55.8 | 84.6 | 924 | 32.42 / 0.8977 | 28.74 / 0.7847 | 27.68 / 0.7400 | 26.52 / 0.7983 | 30.94 / 0.9135 |
| MambaIRV2-lt | 153.4 | 75.6 | 790 | 32.51 / 0.8992 | 28.84 / 0.7878 | 27.75 / 0.7426 | 26.82 / 0.8079 | 31.24 / 0.9182 |
| RDN | 66.0 | 1309.2 | 22271 | 32.47 / 0.8990 | 28.81 / 0.7871 | 27.72 / 0.7419 | 26.61 / 0.8028 | 31.00 / 0.9151 |
| RCAN | 52.2 | 917.6 | 15592 | 32.63 / 0.9002 | 28.87 / 0.7889 | 27.77 / 0.7436 | 26.82 / 0.8087 | 31.22 / 0.9173 |
| CATANet (CVPR'25) | 102.4 | 49.3 | 535 | 32.68 / 0.8998 | 28.90 / 0.7880 | 27.75 / 0.7427 | 26.87 / 0.8081 | 31.31 / 0.9183 |
| ESC (ICCV'25) | 21.9 | 149.2 | 968 | 32.68 / 0.9011 | 28.93 / 0.7902 | 27.80 / 0.7447 | 27.07 / 0.8144 | 31.54 / 0.9207 |
| CATANet-F (Ours) | 110.3 | 56.7 | 941 | 32.69 / 0.9017 | 28.93 / 0.7906 | 27.82 / 0.7444 | 27.05 / 0.8133 | 31.62 / 0.9214 |
| ESC-F (Ours) | 24.6 | 73.4 | 952 | 32.74 / 0.9032 | 28.99 / 0.7931 | 27.86 / 0.7483 | 27.11 / 0.8158 | 31.78 / 0.9233 |

Table 10: Quantitative comparison on benchmark datasets for $\times 4$ lightweight image super-resolution.

## E.6 Light-weight Model Comparison

To demonstrate the effectiveness of our method even under computation-constrained scenarios, we integrate our Factorized Features framework to state-of-the-art lightweight super-resolution models and observe consistent improvement as shown in table 10.

## E.7 Task-Specific Overfitting

To confirm that our **Factorized Features** are not overfitted to a single task, we further apply the same representation to other low-level vision problems such as motion deblurring in table 11 and image dehazing in table 12, without any task-specific modification. The consistent improvements across these tasks demonstrate that our features generalize well and capture transferable visual priors.

| Metric / Method | DeblurGAN-v2 | SRN | DMPHN | SDWNet | MPRNet | MIMO-UNet+ | DeepRFT+ | MAXIM-3S | Stripformer | MSDI-net | Restormer | NAFNet | FFTformer | GRL-B | MLWNet (CVPR'24) | MLWNet-F (Ours) |
|---|---|---|---|---|---|---|---|---|---|---|---|---|---|---|---|---|
| PSNR | 29.55 | 30.26 | 31.20 | 31.26 | 32.66 | 32.45 | 33.23 | 32.86 | 33.08 | 33.28 | 33.57 | 33.69 | 34.21 | 33.93 | 33.83 | **34.40** |
| SSIM | 0.934 | 0.934 | 0.945 | 0.966 | 0.959 | 0.957 | 0.963 | 0.961 | 0.962 | 0.964 | 0.966 | 0.967 | 0.969 | 0.968 | 0.968 | **0.971** |

Table 11: Performance comparison on the GoPro dataset for motion deblurring.

| Metric / Method | DehazeNet | AOD-Net | MSBDN | FFA-Net | AECR-Net | DeHamer | PMNet | DehazeFormer | TaylorFormer | LH-Net | MITNet | PGH²Net (AAAI'24) | ConvIR (TPAMI'24) | ConvIR-F (Ours) |
|---|---|---|---|---|---|---|---|---|---|---|---|---|---|---|
| PSNR | 13.84 | 13.14 | 15.37 | 14.39 | 15.80 | 16.62 | 16.79 | 16.29 | 16.66 | 18.87 | 16.97 | 17.02 | 17.45 | **18.12** |
| SSIM | 0.43 | 0.41 | 0.49 | 0.45 | 0.47 | 0.56 | 0.51 | 0.51 | 0.56 | 0.561 | 0.606 | 0.61 | 0.802 | **0.823** |

Table 12: Performance comparison on the Dense-Haze dataset for image dehazing.

## E.8 Analysis of Model Hyperparameters

We further conduct a comprehensive study on three key hyperparameters of our model: the number of frequency components $N$, the frequency modulation $\alpha$, and the basis transformation $\gamma$ in table 13 and table 14. These factors collectively control the expressiveness and efficiency of the proposed **Factorized Features**.

These results show that our design choices provide a good trade-off between representation flexibility and efficiency, while maintaining stable performance across a wide range of hyperparameter configurations.

| $N$ (Number of Frequency Components) | 1 | 1 | 2 | 6 | 6 | 6 | 6 | 10 |
|---|---|---|---|---|---|---|---|---|
| $\alpha$ | $\{1\}$ | $\{1, 4, 16, 64\}$ | $\{1, 4, 16, 64\}$ | $\{1\}$ | $\{1, 4\}$ | $\{1, 4, 16, 64\}$ | $\{1, 4, 16, 64, 256, 1024\}$ | $\{1, 4, 16, 64\}$ |
| Set5 PSNR | 32.62 | 32.64 | 32.70 | 32.62 | 32.71 | 32.74 | 32.69 | 32.73 |
| Set5 SSIM | 0.9018 | 0.9020 | 0.9029 | 0.9020 | 0.9027 | 0.9032 | 0.9024 | 0.9033 |

Table 13: Ablation on the number of frequency components $N$ and scaling coefficients $\alpha$. We select $N = 6$ for balanced quality and training efficiency, as $N = 10$ yields marginal gains but converges slower.

| Set5 Performance / $\gamma$ | Sawtooth | Sin | Triangular |
|---|---|---|---|
| **PSNR** | 32.71 | 32.69 | 32.71 |
| **SSIM** | 0.9027 | 0.9026 | 0.9023 |

Table 14: Comparison of alternative transformation bases $\gamma$.

# F   Failure Cases

Failure cases typically involve extremely fine textures—details that are absent in the low-resolution (LR) input but appear in the super-resolved (SR) output. However, this has been a shared inherent limitation of super-resolution methods to date. For example, in top row of fig. 6, some white structures are prevent in HR but completely absent in LR.

# G   Future Generalization

For image generation, for example, we can integrate our Factorized Features into the VAE decoder of a diffusion model. For video generation, we can employ our mergeable basis and leverage its properties to enhance structural coherence.

# H   More Visualization

## H.1   Single Image Super-Resolution

Below, we provide more visual comparisons of single-image super-resolution, where FIPER denotes HAT-F.

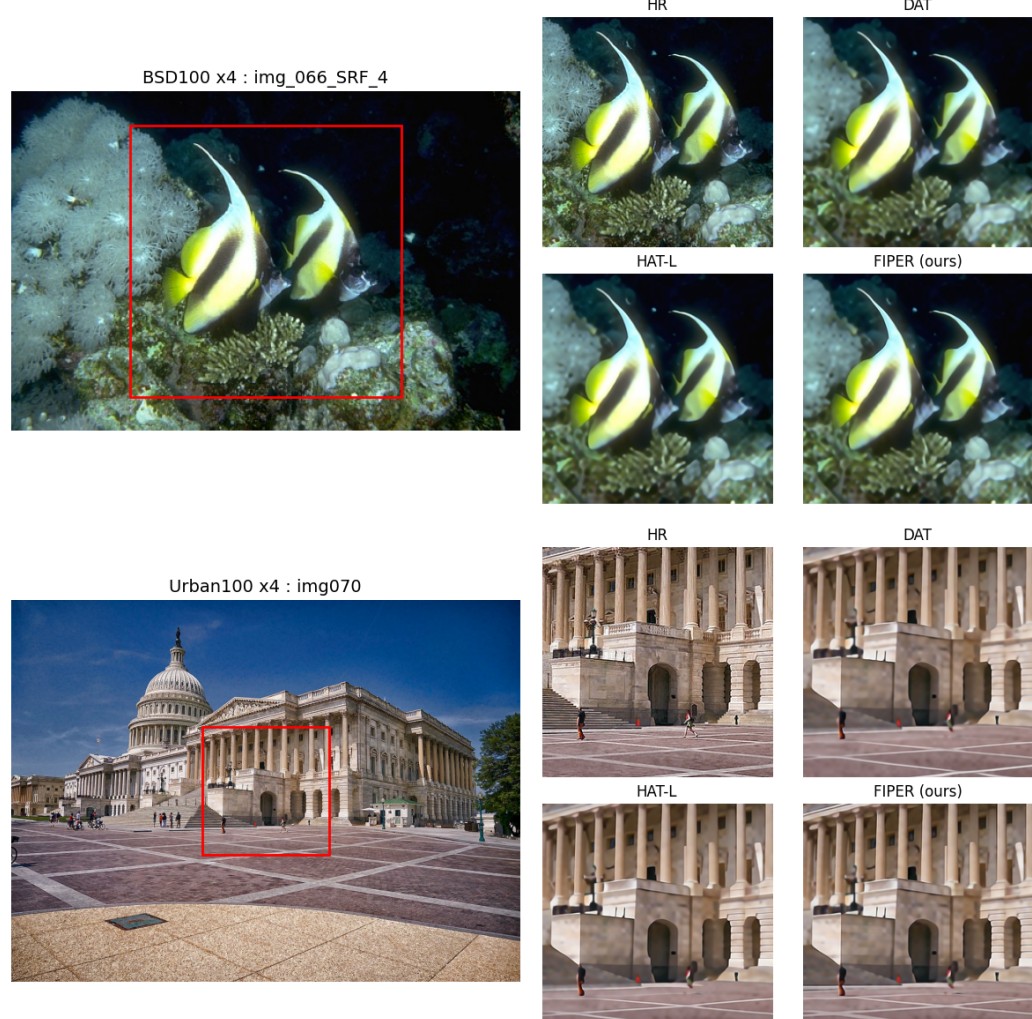

Urban100 x4 : img046

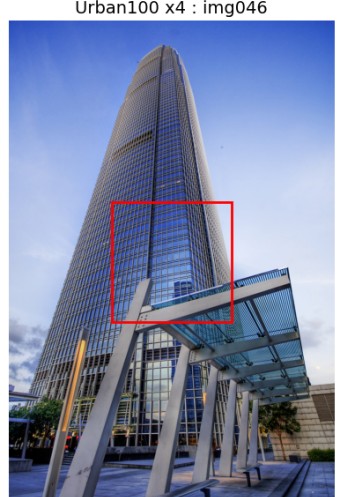

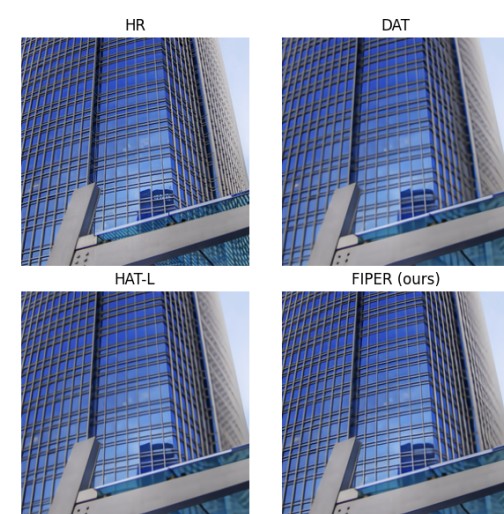

HR          DAT

HAT-L       FIPER (ours)

## H.2 Image Compression

Below, we provide more visual comparisons of image compression. The results are present with TCM-HAT-F and the metrics are PSNR/Bpp.

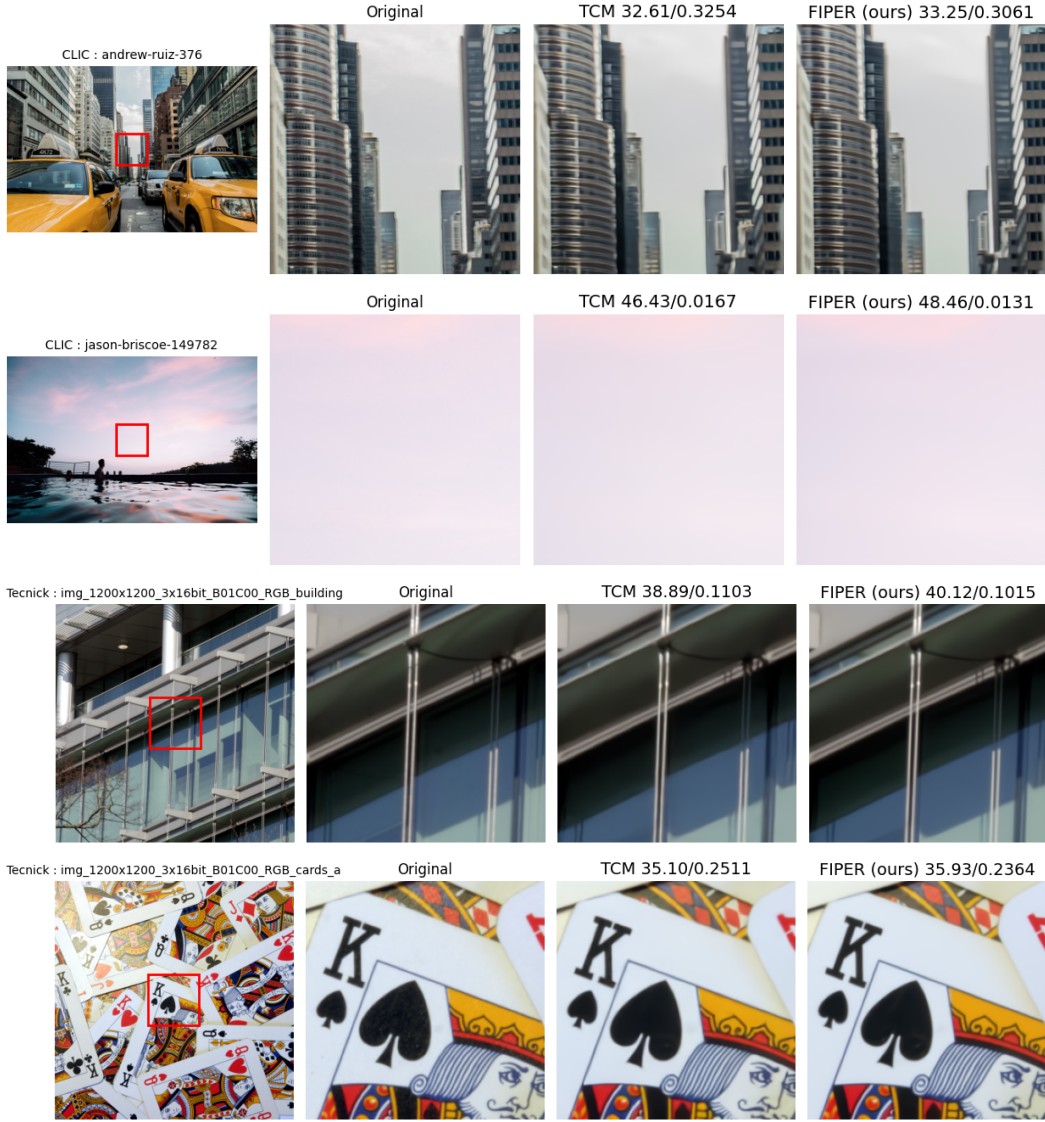

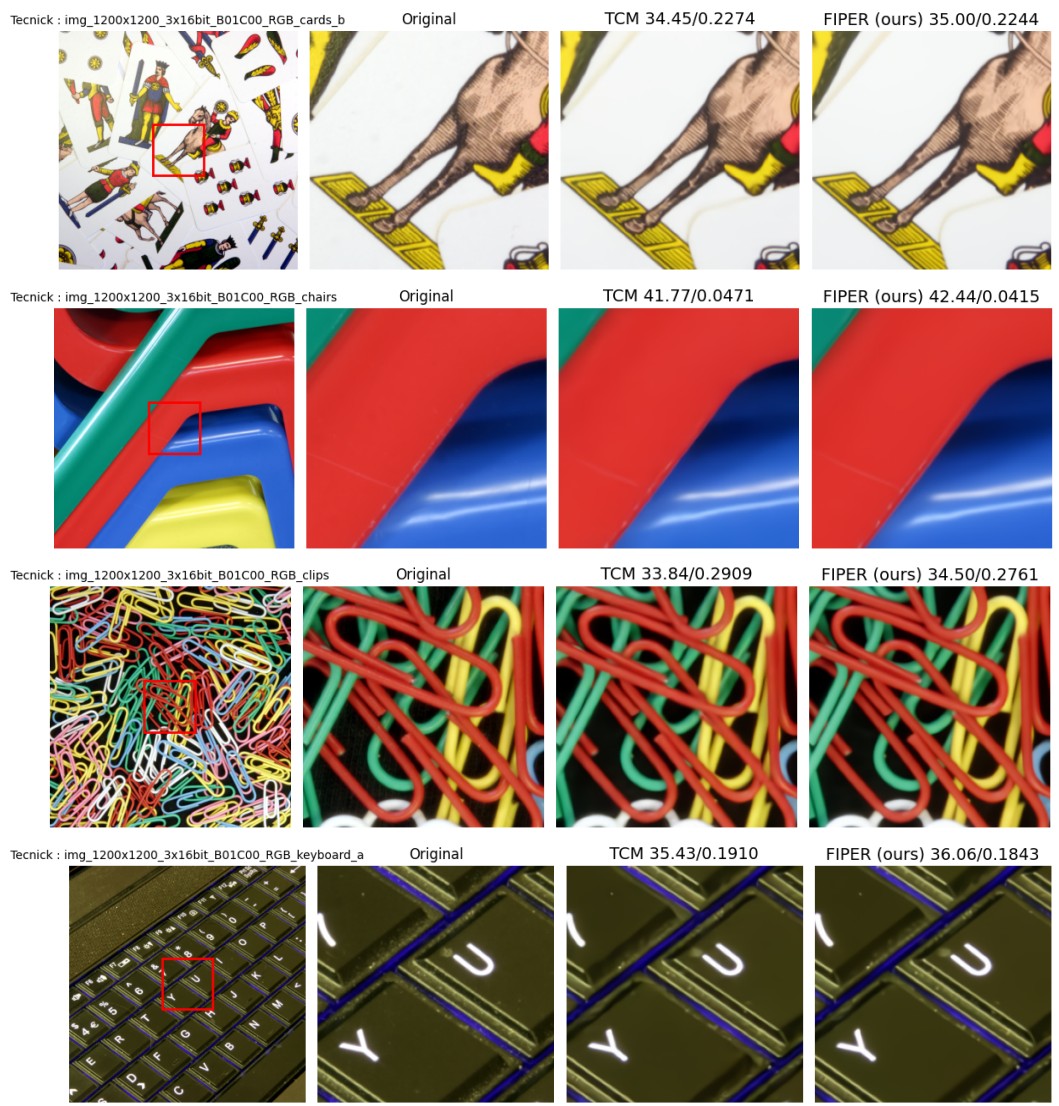

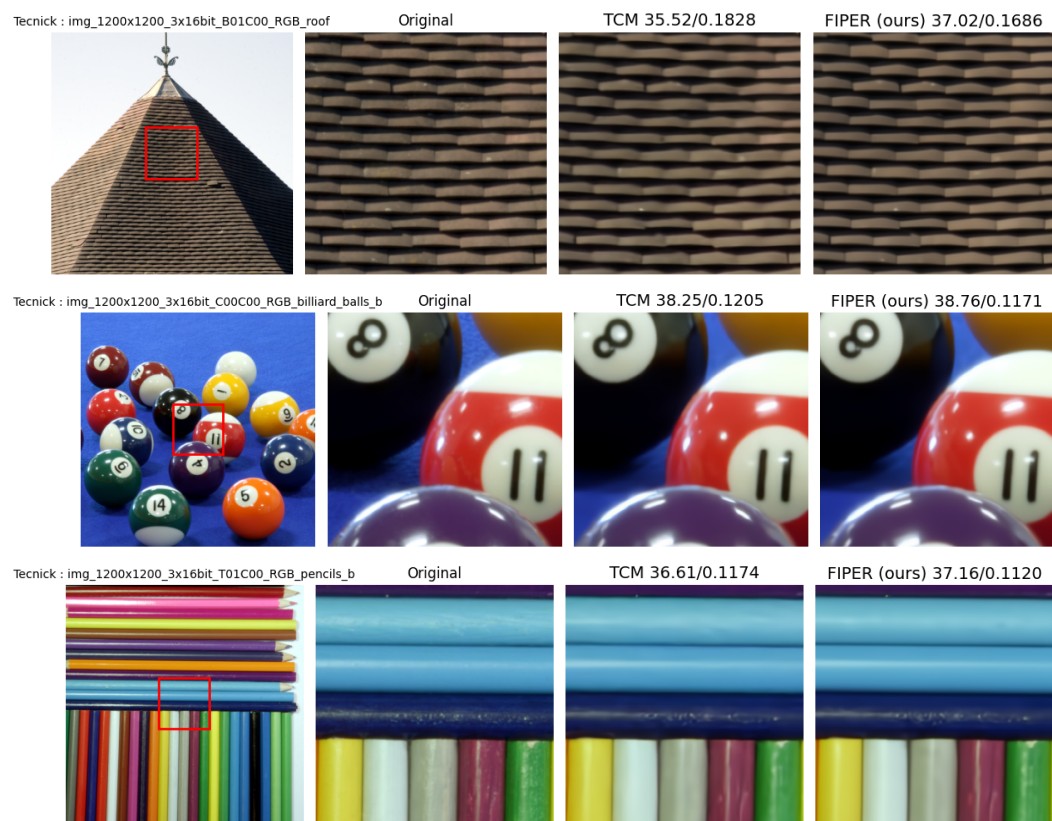

