# OpenReview forum: "FIPER: Factorized Features for Robust Image Super-Resolution and Compression"
_NeurIPS.cc/2025/Conference — NeurIPS 2025 poster_

### Official Review · Reviewer_evQ6 · 2025-06-29

**Clarity:** 2
**Significance:** 3
**Originality:** 3
**Rating:** 5
**Confidence:** 4

**Summary:**

1. The paper proposed a general feature decomposition method that can handle both image super resolution and compression.
2. Instead of using fixed basis as fourier transform and wavelet transform, this paper is proposed to generate image specific basis, which help to improve the high-frequency repeating patterns recovery.
3. Experiments show that the proposed method achieves SOTA performance on multiple benchmarks.

**Questions:**

1. What is z in equation (9)? And what is the R here?
2. The learned coefficients and basis are pixel-wise, but Fourier transform is global. So how to extend equation (2) to (3) is not clear to me.
3. In Fig. 2(b), what is the meaning of "trainable basis settings" here, I thought x and y are fixed.
4. Can authors explain more about the alpha in Figure. (4)?
5. Any visualization of the extracted high-frequency features?
6. How to learn the k? Is it a pure scalar or the output of some conv layers.
7. Figure 5 looks like learning the coefficients and basis in parallel. I do not fully understand why authors claimed "our method derives the coefficient features first"
8. Any Coordinate Transformation tried other than Sawtooth?

**Ethical Concerns:**

["NO or VERY MINOR ethics concerns only"]

**Final Justification:**

Based on the feedback from the authors, all of my concerns are solved and I have updated my scores.

**Limitations:**

Can authors explain more about the following limitation?
Although effective, our method requires further optimization for computation-limited scenarios such as real-time decoding.

**Quality:**

3

**Strengths And Weaknesses:**

1. The experiments are solid. Toy example in Figure 2 shows the advantage of learnable basis.
2. The pipeline is novel, which combines the advantage of both deep network and fixed basis.
3. The description is a bit incoherent, and sometimes hard to follow. Figures (5) lack captions, which need some captions for better understanding.
4. Experiments show that repeating patterns can be better recovered.

---

> ### Author Rebuttal · Authors · 2025-07-31
>
> We sincerely thank the reviewer for recognizing our solid experiments and novel pipeline. We apologize for the clarity issues and address each question comprehensively below:
>
> > **Q1. Notation in Equation (9)**
>
> We apologize for any confusion caused by our notation. We mainly follow the notations and definitions in [1]. In Equation (9): $L = R(\hat{y}) + R(\hat{z}) + \lambda\,D(x, \hat{x})$
>
> - **$\hat{z}$** refers to the **hyperprior**, an auxiliary latent used to estimate the scale parameters for entropy coding, following the standard variational hyperprior framework.
> - **$R(\cdot)$** denotes the **bit-rate cost** of encoding a quantized variable, representing the rate term in the rate–distortion objective.
>
> Thus, $R(\hat{y}) + R(\hat{z})$ measures the total bit cost for both the primary latents and the hyperprior, and $\lambda\, D(x, \hat{x})$ balances this cost against reconstruction quality. We will clarify these definitions in the revised manuscript.
>
> [1] Ballé, Johannes, David Minnen, Saurabh Singh, Sung Jin Hwang, and Nick Johnston. "Variational image compression with a scale hyperprior." ICLR 2018.
>
>
> > **Q2. Pixel-wise vs Global Transform**
>
> Thank you for raising this point. In the classical Fourier representation, each frequency component is applied uniformly across the whole image: a single scalar coefficient multiplies a fixed sinusoidal pattern, so every pixel “sees” the same globally. This is powerful for capturing overall periodicity, but it cannot adapt to local variations in texture or detail.
>
> In our Factorized Features framework, we generalize this in two key ways, trading pure global consistency for **content-aware flexibility** while retaining frequency modeling:
>
> 1. **Learnable, pixel-wise basis maps**
>    Instead of a fixed sinusoid, each basis becomes a full-resolution map that can vary from one pixel to the next, so that regions with different patterns (e.g., smooth sky vs. brick wall) get different frequency accommodations.
>
> 2. **Spatially-varying coefficient maps**
>    Rather than a single scalar per frequency, we produce a coefficient **map** for each basis. This allows each location to choose how strongly it uses each basis, adapting dynamically to local edge strength, texture complexity, or noise.
>
> Together, these two changes turn the rigid global transform into a **hybrid**:
>     - It still captures multi-frequency structure (low and high frequencies) across the image,
>     - But it now **automatically adapts** to fine details and local content via learnable maps.
>
> Why this matters
> - **Local Detail Preservation:** Regions that need more high-frequency emphasis (e.g., sharp edges) can boost those basis maps without affecting smooth regions.
> - **Task Generality:** This flexible decomposition works not only for super-resolution but also for compression (and, in principle, deblurring or denoising), since all these tasks benefit from recovering both broad structures and subtle textures.
>
> In short, we move from “one size fits all” sinusoidal bases and scalar coefficients to a **rich, pixel-wise basis–coefficient decomposition**, which preserves the spirit of Fourier analysis while unlocking powerful local adaptation.
>
> > **Q3. Figure 2(b) Clarification**
>
> We thank the reviewer for this question. In Fig. 2(b), “trainable basis settings” means that we enable gradient updates on the coordinate basis by setting require_grad=True for the (x,y) tensors. In other words, although initialized with fixed grid values, these coordinates become learnable parameters during training (see lines 170–171). We will clarify this point in the revised manuscript.
>
> > **Q4. Alpha (α) in Figure 4**
>
> We can see in Figure 4 with sawtooth transformation $\gamma$ the feature is prompted to learn pattern from different scale in a patch-like manner, e.g. $\gamma_2$ makes the same basis repeat four times in the grid.
> However, only $\gamma$ is not enough for very fine details; therefore, we introduce $\alpha$ and $\psi$. Each basis is forced to accommodate both high-frequency (large $\alpha$) and low-frequency (small $\alpha$) components together with periodic functions $\psi$, i.e., larger $\alpha$ will make $\psi$ oscillate sharper.
> To be more general, sawtooth transformation is to form a patch-like pattern, and $\alpha$ is to model pixel-wise high-frequency details.
>
> > **Q5. High-frequency Feature Visualization**
>
> Note that due to rebuttal guidelines, we can not include images or external links here. We will include visual examples of the extracted high-frequency feature maps in the revised version to illustrate what the network learns.
>
> > **Q6. Learning k**
>
> - **$k$**: Pure scalar hyperparameter in sawtooth transformation in line 162, where we use $\{256, 128, 64, 32,16,8\}$. It is not the output of convolutional layers, found empirically using a validation set and fixed throughout training.
> - **$K$**: The number of $\alpha$. We conduct experiments to validate the setting.
>
> |                |     |     |     |     |     |     |     |     |
> |----------------|-----|-----|-----|-----|-----|-----|-----|-----|
> | **N**  (the number of frequency components)          | $1$     | $1$              | $2$              | $6$     | $6$       | $6$             | $6$                      | $10$            |
> | **$\alpha$**  (modulation frequencies)   | $\{1\}$ | $\{1,4,16,64\}$  | $\{1,4,16,64\}$  | $\{1\}$ | $\{1,4\}$ | $\{1,4,16,64\}$ | $\{1,4,16,64,256,1024\}$ | $\{1,4,16,64\}$ |
> | **Set5 PSNR**  |  32.62  |   32.64          |     32.70        | 32.62   |  32.71    |  32.74          |         32.69            |     32.73       |
> | **Set5 SSIM**  |  0.9018 |   0.9020         |     0.9029       | 0.9020  |  0.9027   |  0.9032         |         0.9024           |     0.9033      |
>
> We will make this explicit in the revised manuscript.
>
> > **Q7. Sequential vs Parallel Processing**
>
> Although the coefficient and basis branches operate in parallel, the main feature $X_{coeff}$ is extracted via the backbone network. The basis transformer then downsamples those features to construct the basis. We will rephrase the caption and accompanying text to clearly separate these two stages.
>
> > **Q8. Alternative Coordinate Transformations**
>
> | Set5 Performance / $\gamma$ | Sawtooth | Sin    | Triangular |
> | --------------------------- | -------- | ---    | ---------- |
> | PSNR                        | 32.71    | 32.69  |  32.71     |
> | SSIM                        | 0.9027   | 0.9026 | 0.9023     |
>
>
> > **L1. Computational Limitations**
>
> Thank you for asking us to clarify this point. By “computation-limited scenarios,” we refer to environments such as on-device real-time decoding where both latency and memory are severely constrained. Although our Factorized Features design already reduces parameters, FLOPs, and peak memory compared to prior methods, the extra steps of basis generation and coefficient fusion still introduce non-negligible overhead under real-time constraints.
>
> However, to demonstrate our method’s effectiveness even under strict computational constraints, we integrated the Factorized Features framework into state-of-the-art lightweight super-resolution models and observed consistent performance gains.
>
> |Method|Latency_ms|#FLOPs_G|#Params_K|Set5_PSNR/SSIM|Set14_PSNR/SSIM|B100_PSNR/SSIM|Urban100_PSNR/SSIM|Manga109_PSNR/SSIM|
> |---|---|---|---|---|---|---|---|---|
> |SwinIR-lt|222.9|63.6|930|32.44/0.8976|28.77/0.7858|27.69/0.7406|26.47/0.7980|30.92/0.9151|
> |ELAN-lt|18.0|54.1|640|32.43/0.8975|28.78/0.7858|27.69/0.7406|26.54/0.7982|30.92/0.9150|
> |OmniSR|22.5|50.9|792|32.49/0.8988|28.78/0.7859|27.71/0.7415|26.64/0.8018|31.02/0.9150|
> |SRFormer-lt|287.2|62.8|873|32.51/0.8988|28.82/0.7872|27.73/0.7422|26.67/0.8032|31.17/0.9165|
> |ATD-lt|189.7|100.1|769|32.63/0.8998|28.89/0.7886|27.79/0.7440|26.97/0.8107|31.48/0.9198|
> |HiT-SRF|82.1|58.0|866|32.55/0.8999|28.87/0.7880|27.75/0.7432|26.80/0.8069|31.26/0.9171|
> |ASID-D8|61.8|49.6†|748|32.57/0.8990|28.89/0.7898|27.78/0.7449|26.89/0.8096|-|
> |MambaIR-lt|55.8|84.6|924|32.42/0.8977|28.74/0.7847|27.68/0.7400|26.52/0.7983|30.94/0.9135|
> |MambaIRV2-lt|153.4|75.6|790|32.51/0.8992|28.84/0.7878|27.75/0.7426|26.82/0.8079|31.24/0.9182|
> |RDN|66.0|1309.2|22271|32.47/0.8990|28.81/0.7871|27.72/0.7419|26.61/0.8028|31.00/0.9151|
> |RCAN|52.2|917.6|15592|32.63/0.9002|28.87/0.7889|27.77/0.7436|26.82/0.8087|31.22/0.9173|
> |CATANet_CVPR25|102.4|49.3|535|32.58/0.8998|28.90/0.7880|27.75/0.7427|26.87/0.8081|31.31/0.9183|
> |ESC_ICCV25|21.9|149.2|968|32.68/0.9011|28.93/0.7902|27.80/0.7447|27.07/0.8144|31.54/0.9207|
> |CATANet-F_Ours|110.3|56.7|941|32.69/0.9017|28.93/0.7906|27.82/0.7444|27.05/0.8133|31.62/0.9214|
> |ESC-F_Ours|24.6|73.4|952|32.74/0.9032|28.99/0.7931|27.86/0.7483|27.11/0.8158|31.78/0.9233|
>
>
> > **S&W3. Figure 5 Caption**
>
> Thank you for highlighting this point. We agree that the presentation around Figure 5 can be made more coherent, and that explicit captions will help readers follow the workflow. In the revised manuscript, we will:
> - **Clarify the narrative:** Reorganize the text in Sections 4.2 and 4.3 to present the coefficient and basis extraction steps in a clear, sequential manner.
> - **Add detailed captions:**
> > This figure illustrates how Factorized Features are used in Super-Resolution and Image Compression. (a) Given a low-resolution input image, we first extract $X_{coeff}$ feature with Coefficient Backbone. Next, we generate the basis and coefficient with Basis Swin Transformer and convolution layers, respectively, from the same $X_{coeff}$. Finally, the prediction is reconstructed by Factorized Features Reconstruction. (b) To decrease distortion in image compression, we replace the synthesis transform of the traditional learned image compression pipeline with (a) by aligning spatial resolution and latent channels.
>
>
> These changes will ensure the description is coherent and that Figure 5 stands on its own as a self-contained illustration of our method.

---

> > ### Comment · Reviewer_evQ6 · 2025-08-04
> >
> > Thanks for the feedback from the authors. I think all my concerns are solved.

---

> ### Author Response · Authors · 2025-08-05
>
> Thank you again for your detailed feedback and for confirming that all concerns have been addressed. We appreciate your time and judgment.

---

> ### Comment · Area_Chair_4crZ · 2025-08-05
>
> Dear Reviewer evQ6,
>
> Thank you for your effort in the review process, including the response to authors' feedback. If you believe the discussion with authors can be concluded, please submit the Mandatory Acknowledgement as required by the program of the conference.
>
> Thank you,
>
> AC

---

### Official Review · Reviewer_YrvH · 2025-07-02

**Clarity:** 4
**Significance:** 3
**Originality:** 3
**Rating:** 4
**Confidence:** 3

**Summary:**

This paper proposes "Factorized Features," a unified representation for low-level vision tasks, specifically demonstrating its effectiveness in Single Image Super-Resolution (SISR) and Image Compression. Unlike prior work focusing on network architecture, their approach uses a learned basis-coefficient decomposition with explicit frequency formulation to capture structural and multi-scale features. This representation is designed to generalize broadly, replacing simple feature maps in existing models. The authors further optimize image compression by leveraging a mergeable-basis property for multi-frame scenarios. Extensive experiments show state-of-the-art performance, with significant PSNR gains in SR and BD-rate reduction in image compression.

**Questions:**

1. For "broad generalizability," the paper only evaluates on two low-level vision tasks (SISR and image compression). Expanding to other tasks like denoising, deblurring, or inpainting would further strengthen the claim.

2. The Lack of ablation:
  2.1 Could a more comprehensive ablation study be provided to quantify the individual contribution of each innovative component (e.g., presence/absence of spatially variant coefficients, learned non-uniform bases, coordinate transformation, and multi-frequency modulation) to the final performance? This would be crucial for understanding which elements are the key drivers behind the observed performance improvements.
2.2 Could an ablation study on the choice of N (the number of frequency components) be provided? How do different values of N affect the model's performance, complexity, and its ability to capture fine image details? Is there an optimal value of N, or does it vary depending on the specific task?

**Ethical Concerns:**

["NO or VERY MINOR ethics concerns only"]

**Final Justification:**

This paper presents a novel and effective framework for content-aware basis learning, with demonstrated benefits in low-level vision tasks. The authors have provided a thorough and convincing rebuttal, addressing all raised concerns with additional explanations, ablation studies, and extended evaluations on diverse tasks such as motion deblurring and dehazing.

**Limitations:**

See Questions.

**Quality:**

4

**Strengths And Weaknesses:**

Strength

1. The concept of "Factorized Features" as a unified representation for diverse low-level vision tasks like SR and compression is a significant and novel contribution. This approach moves beyond purely architectural improvements to focus on fundamental image content representation.
2. By replacing simple feature maps with Factorized Features, the paper successfully demonstrates the potential for broad generalizability across different low-level vision tasks and existing backbones.


Weaknesses:

1. While "Learned Non-uniform Basis" is introduced, the mechanism for generating the entire basis map  for better fitting and local reconstruction needs more in-depth explanation beyond just "networks to generate."

2. While a "mergeable-basis property" is mentioned for multi-image compression, the explanation (Eq. 10 and related text) is brief and could be expanded to clarify the practical benefits and challenges in this setup.

---

> ### Author Rebuttal · Authors · 2025-07-31
>
> We sincerely thank the reviewer for recognizing our work as "excellent" in quality and clarity, and for acknowledging our "significant and novel contribution." We address all concerns comprehensively below:
>
> > **W1. Learned Non-uniform Basis Mechanism**
>
> In lines 89-100 of the Supplement, we describe implementation details of how the basis is generated. To be more clear, given an image, we extract a feature $X_{coeff}$ by Coefficient Backbone, and then the Basis Transformer is applied to refine the feature with downsampling for final basis generation. After the reconstruction with the coefficient and basis, the prediction is used to compute the loss and subsequently optimize the network. With gradient descent, when the Basis Transformer is optimized, the basis is optimized simultaneously, i.e., the models are trained to adapt to content-aware basis, so the basis is not calculated by heuristic rules. This design enables **content-aware basis generation**: smooth regions get low-frequency bases while textured areas receive high-frequency ones. We will expand this explanation with architectural diagrams in the revision.
>
>
> > **W2. Mergeable-basis Property Explanation**
>
> Thank you for this valuable suggestion. We will expand the discussion around Eq. 10 in the main text to clarify both the benefits and limitations of our mergeable-basis approach. Specifically:
>
> As detailed in lines 133–141 of the Supplementary, our merged basis exploits the observation that different images often share common frequency components. By learning a single, shared set of basis functions, we can leverage the frequently appearing similar or repetitive patterns across images to greatly reduce coding redundancy in multi-image compression. In practice, this means fewer total parameters and lower computational cost required when compressing an image collection. We also note the trade-off: when images are highly diverse, forcing them into one shared basis can increase reconstruction error (as seen by the quality drop for larger M in Fig. 7). Still, this content-aware information redundancy exploitation enables a significant reduction in resource usage, and we will make these points explicit in the revised manuscript.
>
> > **Q1. Evaluation on More Tasks**
>
> We acknowledge this limitation and have extended our evaluation:
> #### GoPro Dataset (Motion Deblurring)
> | Metric / Method  | DeblurGAN-v2 | SRN  | DMPHN | SDWNet | MPRNet | MIMO-UNet+ | DeepRFT+ | MAXIM-3S | Stripformer | MSDI-net | Restormer | NAFNet | FFTformer | GRL-B | MLWNet (CVPR'24) | MLWNet-F |
> |------------------|--------------|------|-------|--------|--------|------------|----------|----------|-------------|----------|-----------|--------|-----------|-------| ---------------- | -------- |
> | **PSNR**         | 29.55        | 30.26| 31.20 | 31.26  | 32.66  | 32.45      | 33.23    | 32.86    | 33.08       | 33.28    | 33.57     | 33.69  | 34.21     | 33.93 | 33.83            | 34.4     |
> | **SSIM**         | 0.934        | 0.934| 0.945 | 0.966  | 0.959  | 0.957      | 0.963    | 0.961    | 0.962       | 0.964    | 0.966     | 0.967  | 0.969     | 0.968 | 0.968            | 0.971    |
> #### Dense-Haze Dataset (Dehazing)
> | Metric / Method | DehazeNet | AOD-Net | MSBDN | FFA-Net | AECR-Net | DeHamer | PMNet | DehazeFormer | TaylorFormer | LH-Net | MITNet | PGH^2Net (AAAI'24) | ConvIR (TPAMI'24) | ConvIR-F|
> |-----------------|-----------|---------|-------|---------|----------|---------|-------|--------------|--------------|--------|--------|-----------         | ---               | --      |
> | **PSNR**        | 13.84     | 13.14   | 15.37 | 14.39   | 15.80    | 16.62   | 16.79 | 16.29        | 16.66        | 18.87  | 16.97  | 17.02              | 17.45             | 18.12   |
> | **SSIM**        | 0.43      | 0.41    | 0.49  | 0.45    | 0.47     | 0.56    | 0.51  | 0.51         | 0.56         | 0.561  | 0.606  | 0.61               | 0.802             | 0.823   |
>
> Our framework clearly generalizes across diverse low-level vision tasks. Due to time constraints in the rebuttal phase, we have focused on these two additional benchmarks, but we will include further evaluations on denoising, deraining, and other tasks in the revised manuscript.
>
> Futhermore, to demonstrate our method’s broader effectiveness, we integrated the Factorized Features framework into state-of-the-art lightweight super-resolution models and observed consistent performance gains.
>
> |Method|Latency_ms|#FLOPs_G|#Params_K|Set5_PSNR/SSIM|Set14_PSNR/SSIM|B100_PSNR/SSIM|Urban100_PSNR/SSIM|Manga109_PSNR/SSIM|
> |---|---|---|---|---|---|---|---|---|
> |SwinIR-lt|222.9|63.6|930|32.44/0.8976|28.77/0.7858|27.69/0.7406|26.47/0.7980|30.92/0.9151|
> |ELAN-lt|18.0|54.1|640|32.43/0.8975|28.78/0.7858|27.69/0.7406|26.54/0.7982|30.92/0.9150|
> |OmniSR|22.5|50.9|792|32.49/0.8988|28.78/0.7859|27.71/0.7415|26.64/0.8018|31.02/0.9150|
> |SRFormer-lt|287.2|62.8|873|32.51/0.8988|28.82/0.7872|27.73/0.7422|26.67/0.8032|31.17/0.9165|
> |ATD-lt|189.7|100.1|769|32.63/0.8998|28.89/0.7886|27.79/0.7440|26.97/0.8107|31.48/0.9198|
> |HiT-SRF|82.1|58.0|866|32.55/0.8999|28.87/0.7880|27.75/0.7432|26.80/0.8069|31.26/0.9171|
> |ASID-D8|61.8|49.6†|748|32.57/0.8990|28.89/0.7898|27.78/0.7449|26.89/0.8096|-|
> |MambaIR-lt|55.8|84.6|924|32.42/0.8977|28.74/0.7847|27.68/0.7400|26.52/0.7983|30.94/0.9135|
> |MambaIRV2-lt|153.4|75.6|790|32.51/0.8992|28.84/0.7878|27.75/0.7426|26.82/0.8079|31.24/0.9182|
> |RDN|66.0|1309.2|22271|32.47/0.8990|28.81/0.7871|27.72/0.7419|26.61/0.8028|31.00/0.9151|
> |RCAN|52.2|917.6|15592|32.63/0.9002|28.87/0.7889|27.77/0.7436|26.82/0.8087|31.22/0.9173|
> |CATANet[1] (CVPR'25)|102.4|49.3|535|32.58/0.8998|28.90/0.7880|27.75/0.7427|26.87/0.8081|31.31/0.9183|
> |ESC[2] (ICCV'25)|21.9|149.2|968|32.68/0.9011|28.93/0.7902|27.80/0.7447|27.07/0.8144|31.54/0.9207|
> |CATANet-F (Ours)|110.3|56.7|941|32.69/0.9017|28.93/0.7906|27.82/0.7444|27.05/0.8133|31.62/0.9214|
> |ESC-F (Ours)|24.6|73.4|952|32.74/0.9032|28.99/0.7931|27.86/0.7483|27.11/0.8158|31.78/0.9233|
>
> [1] Liu, X., Liu, J., Tang, J., & Wu, G. (2025). CATANet: Efficient Content-Aware Token Aggregation for Lightweight Image Super-Resolution. In Proceedings of the Computer Vision and Pattern Recognition Conference (pp. 17902-17912).
>
> [2] Lee, D., Yun, S., & Ro, Y. (2025). Emulating Self-attention with Convolution for Efficient Image Super-Resolution. arXiv preprint arXiv:2503.06671.
>
>
>
>
> > **Q2. Comprehensive Component Ablation**
>
> The experiment is conducted with ESC-F as in Q1.
>
> #### Comparison of multi-frequency modulation and the number of frequency components
>
> |                |     |     |     |     |     |     |     |     |
> |----------------|-----|-----|-----|-----|-----|-----|-----|-----|
> | **N**  (the number of frequency components)          | $1$     | $1$              | $2$              | $6$     | $6$       | $6$             | $6$                      | $10$            |
> | **$\alpha$**  (modulation frequencies)  | $\{1\}$ | $\{1,4,16,64\}$  | $\{1,4,16,64\}$  | $\{1\}$ | $\{1,4\}$ | $\{1,4,16,64\}$ | $\{1,4,16,64,256,1024\}$ | $\{1,4,16,64\}$ |
> | **Set5 PSNR**  |  32.62  |   32.64          |     32.70        | 32.62   |  32.71    |  32.74          |         32.69            |     32.73       |
> | **Set5 SSIM**  |  0.9018 |   0.9020         |     0.9029       | 0.9020  |  0.9027   |  0.9032         |         0.9024           |     0.9033      |
>
> #### Comparison of  Coordinate Transformation
>
> | Set5 Performance / $\gamma$ | Sawtooth | Sin    | Triangular |
> | --------------------------- | -------- | ---    | ---------- |
> | PSNR                        | 32.71    | 32.69  |  32.71     |
> | SSIM                        | 0.9027   | 0.9026 | 0.9023     |
>
> We choose $N=6$ for better quality and training efficiency ($N=10$ converges slower). Note that due to time and resource constraint we omit spatial variant coefficients since it is quite intuitive and also learned non-uniform bases since it is discussed in Figure 2.

---

> > ### Comment · Reviewer_YrvH · 2025-08-04
> >
> > Thanks for your detailed response!  Overall, these results have mostly addressed my concerns. I will increase my score.

---

> ### Author Response · Authors · 2025-08-04
>
> Thank you for your kind feedback! We are happy to hear that our responses addressed most of your concerns. Please do not hesitate to point out any remaining questions we might have missed.

---

### Official Review · Reviewer_Wt2E · 2025-07-02

**Clarity:** 3
**Significance:** 3
**Originality:** 3
**Rating:** 4
**Confidence:** 4

**Summary:**

FIPER introduces factorized features, which can be used for low-level vision tasks such as super-resolution and image compression. Unlike previous works, FIPER decomposes features into basis and coefficients while explicitly modelling frequency components. It replaces standard feature maps and demonstrates strong generalization across both tasks. Experiments show significant performance improvement for both super-resolution and image compression tasks.

**Questions:**

Could you report results (accuracy and runtime) with a lightweight CNN or Mobile-ViT backbone to demonstrate wider applicability?

Why choose N = 6 coefficient–basis pairs and alpha ∈ {1, 4, 16, 64}?  An ablation on N and frequency counts would help

Figure 3 lacks clarity, as the results for the toy example appear identical across all cases. Can you shed more light on it?

What would be the effect if, instead of applying the coordinate transformation \gamma to x in Equation 6, we use a sine or cosine transformation?

**Ethical Concerns:**

["NO or VERY MINOR ethics concerns only"]

**Final Justification:**

The authors have addressed my questions in the rebuttal, particularly regarding the extension of their approach to other low-level vision tasks and the demonstration of its effectiveness with lightweight models. Therefore, I am inclined to recommend acceptance.

**Limitations:**

Please see weaknesses and questions

**Paper Formatting Concerns:**

Kindly change fig. and sec to Fig. and Sec. respectively
Improve the readability of Fig.2 and Fig. 3

**Quality:**

3

**Strengths And Weaknesses:**

Strengths:
The core contribution, leveraging a frequency-aware decomposition (basis × coefficient) across both super-resolution and image compression, is well motivated and novel.

The paper reports significant improvements over prior SOTA on multiple SR benchmarks (e.g., Set5, Urban100) and achieves a 21% BD-Rate gain on image compression compared to VTM, outperforming several recent methods like TCM and ELIC.


Weaknesses:

The method requires pre-trained strong backbones (e.g., HAT, DAT). A comparison with lightweight SR/IC methods would improve practical relevance.

The paper presents Factorized Features as a “unified representation for low-level vision,” yet experiments cover only 4× SR and image compression; no evidence is given for de-blurring, denoising, and other low-level vision tasks.

---

> ### Author Rebuttal · Authors · 2025-07-31
>
> We sincerely thank the reviewer for their positive assessment and constructive feedback. We are encouraged that they find our frequency-aware decomposition "well motivated and novel" and acknowledge our significant improvements (21% BD-Rate gain). We address each concern below:
>
> > **W1. Lightweight Backbone Evaluation**
>
> Thank you for this important suggestion. We have conducted additional experiments with lightweight backbones and observed that Factorized Features provides consistent gains even in resource-constrained settings: lightweight models can also benefit from our Factorized Features.
> | Method          | Latency (ms) | #FLOPs (G) | #Params (K) | Set5 PSNR / SSIM | Set14 PSNR / SSIM | B100 PSNR / SSIM | Urban100 PSNR / SSIM | Manga109 PSNR / SSIM |
> |-----------------|:------------:|:----------:|:-----------:|:----------------:|:-----------------:|:----------------:|:--------------------:|:--------------------:|
> | SwinIR-lt       | 222.9        | 63.6       | 930         | 32.44 / 0.8976   | 28.77 / 0.7858    | 27.69 / 0.7406   | 26.47 / 0.7980       | 30.92 / 0.9151       |
> | ELAN-lt         | 18.0         | 54.1       | 640         | 32.43 / 0.8975   | 28.78 / 0.7858    | 27.69 / 0.7406   | 26.54 / 0.7982       | 30.92 / 0.9150       |
> | OmniSR          | 22.5         | 50.9       | 792         | 32.49 / 0.8988   | 28.78 / 0.7859    | 27.71 / 0.7415   | 26.64 / 0.8018       | 31.02 / 0.9150       |
> | SRFormer-lt     | 287.2        | 62.8       | 873         | 32.51 / 0.8988   | 28.82 / 0.7872    | 27.73 / 0.7422   | 26.67 / 0.8032       | 31.17 / 0.9165       |
> | ATD-lt          | 189.7        | 100.1      | 769         | 32.63 / 0.8998   | 28.89 / 0.7886    | 27.79 / 0.7440   | 26.97 / 0.8107       | 31.48 / 0.9198       |
> | HiT-SRF         | 82.1         | 58.0       | 866         | 32.55 / 0.8999   | 28.87 / 0.7880    | 27.75 / 0.7432   | 26.80 / 0.8069       | 31.26 / 0.9171       |
> | ASID-D8         | 61.8         | 49.6†      | 748         | 32.57 / 0.8990   | 28.89 / 0.7898    | 27.78 / 0.7449   | 26.89 / 0.8096       | –                    |
> | MambaIR-lt      | 55.8         | 84.6       | 924         | 32.42 / 0.8977   | 28.74 / 0.7847    | 27.68 / 0.7400   | 26.52 / 0.7983       | 30.94 / 0.9135       |
> | MambaIRV2-lt    | 153.4        | 75.6       | 790         | 32.51 / 0.8992   | 28.84 / 0.7878    | 27.75 / 0.7426   | 26.82 / 0.8079       | 31.24 / 0.9182       |
> | RDN             | 66.0         | 1309.2     | 22271       | 32.47 / 0.8990   | 28.81 / 0.7871    | 27.72 / 0.7419   | 26.61 / 0.8028       | 31.00 / 0.9151       |
> | RCAN            | 52.2         | 917.6      | 15592       | 32.63 / 0.9002   | 28.87 / 0.7889    | 27.77 / 0.7436   | 26.82 / 0.8087       | 31.22 / 0.9173       |
> | CATANet[1] (CVPR'25) | 102.4      | 49.3       | 535         | 32.58 / 0.8998   | 28.90 / 0.7880    | 27.75 / 0.7427   | 26.87 / 0.8081       | 31.31 / 0.9183       |
> | ESC[2] (ICCV'25)     | 21.9       | 149.2      | 968         | 32.68 / 0.9011   | 28.93 / 0.7902    | 27.80 / 0.7447   | 27.07 / 0.8144       | 31.54 / 0.9207       |
> | CATANet-F (Ours)  | 110.3      | 56.7       | 941         | 32.69 / 0.9017   | 28.93 / 0.7906    | 27.82 / 0.7444   | 27.05 / 0.8133       | 31.62 / 0.9214       |
> | ESC-F (Ours)      | 24.6       | 73.4       | 952         | 32.74 / 0.9032   | 28.99 / 0.7931    | 27.86 / 0.7483   | 27.11 / 0.8158       | 31.78 / 0.9233       |
>
>
> Due to time and resource constraints, we have focused on these initial evaluations on super-resolution; comparisons with lightweight image compression methods[3][4] will be included in the final version. We will add this analysis to the paper.
>
> [1] Liu, X., Liu, J., Tang, J., & Wu, G. (2025). CATANet: Efficient Content-Aware Token Aggregation for Lightweight Image Super-Resolution. In Proceedings of the Computer Vision and Pattern Recognition Conference (pp. 17902-17912).
>
> [2] Lee, D., Yun, S., & Ro, Y. (2025). Emulating Self-attention with Convolution for Efficient Image Super-Resolution. arXiv preprint arXiv:2503.06671.
>
> [3] Bao, Y., Tan, W., Jia, C., Li, M., Liang, Y., & Tian, Y. (2025). ShiftLIC: Lightweight Learned Image Compression with Spatial-Channel Shift Operations. IEEE Transactions on Circuits and Systems for Video Technology.
>
> [4] Wang, S., Cheng, Z., Feng, D., Lu, G., Song, L., & Zhang, W. (2024, December). Asymllic: Asymmetric lightweight learned image compression. In 2024 IEEE International Conference on Visual Communications and Image Processing (VCIP) (pp. 1-5). IEEE.
>
>
> > **W2. Limited Task Coverage**
>
> We acknowledge this limitation and have extended our evaluation:
> #### GoPro Dataset (Motion Deblurring)
> | Metric / Method  | DeblurGAN-v2 | SRN  | DMPHN | SDWNet | MPRNet | MIMO-UNet+ | DeepRFT+ | MAXIM-3S | Stripformer | MSDI-net | Restormer | NAFNet | FFTformer | GRL-B | MLWNet (CVPR'24) | MLWNet-F |
> |------------------|--------------|------|-------|--------|--------|------------|----------|----------|-------------|----------|-----------|--------|-----------|-------| ---------------- | -------- |
> | **PSNR**         | 29.55        | 30.26| 31.20 | 31.26  | 32.66  | 32.45      | 33.23    | 32.86    | 33.08       | 33.28    | 33.57     | 33.69  | 34.21     | 33.93 | 33.83            | 34.4     |
> | **SSIM**         | 0.934        | 0.934| 0.945 | 0.966  | 0.959  | 0.957      | 0.963    | 0.961    | 0.962       | 0.964    | 0.966     | 0.967  | 0.969     | 0.968 | 0.968            | 0.971    |
> #### Dense-Haze Dataset (Dehazing)
> | Metric / Method | DehazeNet | AOD-Net | MSBDN | FFA-Net | AECR-Net | DeHamer | PMNet | DehazeFormer | TaylorFormer | LH-Net | MITNet | PGH^2Net (AAAI'24) | ConvIR (TPAMI'24) | ConvIR-F|
> |-----------------|-----------|---------|-------|---------|----------|---------|-------|--------------|--------------|--------|--------|-----------         | ---               | --      |
> | **PSNR**        | 13.84     | 13.14   | 15.37 | 14.39   | 15.80    | 16.62   | 16.79 | 16.29        | 16.66        | 18.87  | 16.97  | 17.02              | 17.45             | 18.12   |
> | **SSIM**        | 0.43      | 0.41    | 0.49  | 0.45    | 0.47     | 0.56    | 0.51  | 0.51         | 0.56         | 0.561  | 0.606  | 0.61               | 0.802             | 0.823   |
>
> Our framework clearly generalizes across diverse low-level vision tasks. Due to time constraints in the rebuttal phase, we have focused on these two additional benchmarks, but we will include further evaluations on denoising, deraining, and other tasks in the revised manuscript.
>
> > **Q1. Runtime with Lightweight Backbones**
>
> Please see W1. Note that ESC is convolution-based while CATANet is a hybrid transformer.
>
> > **Q2. Hyperparameter Choices (N=6, α∈{1,4,16,64})**
>
>
> |                |     |     |     |     |     |     |     |     |
> |----------------|-----|-----|-----|-----|-----|-----|-----|-----|
> | **N**          | $1$     | $1$              | $2$              | $6$     | $6$       | $6$             | $6$                      | $10$            |
> | **$\alpha$**   | $\{1\}$ | $\{1,4,16,64\}$  | $\{1,4,16,64\}$  | $\{1\}$ | $\{1,4\}$ | $\{1,4,16,64\}$ | $\{1,4,16,64,256,1024\}$ | $\{1,4,16,64\}$ |
> | **Set5 PSNR**  |  32.62  |   32.64          |     32.70        | 32.62   |  32.71    |  32.74          |         32.69            |     32.73       |
> | **Set5 SSIM**  |  0.9018 |   0.9020         |     0.9029       | 0.9020  |  0.9027   |  0.9032         |         0.9024           |     0.9033      |
>
> We choose $N=6$ for better quality and training efficiency ($N=10$ converges slower).
>
>
> > **Q3. Figure 3 Clarity**
>
> Figure 3 shows the difference between vanilla fields, coordinate transformation, and multi-frequency modulation. Specifically, 3.a is a vanilla basis-coefficient field, and 3.b adds sawtooth transformation. We can see in 3.b that the coordinate transformation explicitly models a patch-like pattern, e.g., the second plot from the top is divided into two periods. By enforcing such frequency components, the models can decompose the signal effectively.
>
> Next, 3.d is our full formulation, while 3.c has no $\alpha$. Intuitively, from 3.b to 3.c, applying $\psi$ (we use $cos$ here) should not have much effect on the performance since we do not have any constraint on these learnable bases. With the introduction of $\alpha$, the models are forced to attend to signal components of different frequencies.
> Note that due to rebuttal guidelines, we can not include images or external links here; we will adjust the graph layout and font for better clarity.
>
> > **Q4. Alternative Transformations**
>
> The experiment is conducted with ESC-F as in W1.
>
> | Set5 Performance / $\gamma$ | Sawtooth | Sin    | Triangular |
> | --------------------------- | -------- | ---    | ---------- |
> | PSNR                        | 32.71    | 32.69  |  32.71     |
> | SSIM                        | 0.9027   | 0.9026 | 0.9023     |

---

> ### Comment · Area_Chair_4crZ · 2025-08-05
> **Please respond to authors' rebuttal before Aug. 6 (AOE)**
>
> Dear Reviewer Wt2E,
>
> This is a reminder that the author-reviewer discussion period is ending soon on Aug. 6 (AOE), and you have not yet responded to authors' rebuttal. Please read authors' rebuttal as soon as possible, and engage in any necessary discussions, and consider if you would like to update your review and score. Please at least submit the Mandatory Acknowledgement as a sign that you have completed this task.
>
> Thank you for your service in the review process.
>
> AC

---

### Official Review · Reviewer_yir9 · 2025-07-03

**Clarity:** 3
**Significance:** 3
**Originality:** 3
**Rating:** 4
**Confidence:** 4

**Summary:**

This paper is interesting. The authors proposes a 100M network to estimate the best bases and coefficients for feature representation for SR and compression tasks. Experiemental results show obvious improvment in terms of repetitive textures and fine details.

**Questions:**

Failure cases can be given in the paper.

**Ethical Concerns:**

["NO or VERY MINOR ethics concerns only"]

**Limitations:**

Can the Factorized Features be overfitted for certain tasks? i.e. task specific.

**Quality:**

3

**Strengths And Weaknesses:**

A new network design for base and coefficient estimation.
Can this network be applied to other tasks, such as image /video generations?

Strengths
1. Unified Representation for Multiple Tasks: Factorized Features generalize across SISR and Image Compression, addressing core challenges (fine detail recovery, structural preservation) with a single framework, unlike task-specific prior methods.
2. Explicit Frequency and Structural Modeling: By combining basis-coefficient decomposition, spatially variant coefficients, learned non-uniform bases, and multi-frequency modulation, the method effectively captures both low-frequency global structures and high-frequency local details, overcoming limitations of fixed sinusoidal bases (e.g., poor flexibility) and pure network architecture optimizations (e.g., neglect of frequency patterns).
3. Optimized Compression Pipeline: The "mergeable-basis" property enables consolidating shared structures in multi-frame compression, reducing redundancy and improving efficiency.
4. Strong Empirical Performance: Outperforms SOTA methods on benchmarks:
In SR, achieves higher PSNR/SSIM across datasets (e.g., Set5, Urban100).
In compression, delivers significant BD-rate reductions (e.g., -21.09% vs. VTM for single-image, -20.97% for multi-image with M=24).
Cross-Task Prior Integration: Leverages pre-trained SR models as priors for image compression, enhancing information recovery.

Weaknesses
1. Computational Efficiency: The method requires optimization for computation-constrained scenarios (e.g., real-time decoding) due to higher model complexity (e.g., larger parameters and MACs compared to some baselines, as shown in Table 1 and 2).
2. Lack of Semantic Information: The framework does not incorporate semantic understanding, which could limit performance in tasks requiring context-aware detail preservation.
3. Scalability Challenges in Multi-Frame Compression: While merging bases improves efficiency, distortion increases slightly as the number of merged frames (M) grows, stretching basis capacity.
4. Dependence on Backbone Architecture: Performance gains rely on integrating with strong backbones (e.g., HAT, DAT), which may limit applicability to lightweight models.

---

> ### Author Rebuttal · Authors · 2025-07-31
>
> We sincerely thank the reviewer for their thorough and constructive feedback. We are encouraged that they find our paper "interesting" and acknowledge our contributions, including unified representation, explicit frequency modeling, optimized compression pipeline, and strong empirical performance. Below, we address each concern:
>
>
>
> > **W1. Computational Efficiency**
>
> We acknowledge this important practical concern. In Table 1, only the parameter count of our HAT-F is slightly more than its counterpart, HAT-L; the MAC and Memory footprint of all our models are lower than their counterparts. In Table 2, our TCM-HAT-F-multi, though with slightly more parameters, actually achieves competitive encoding time and even the lowest decoding time of all models.
> Furthermore, to demonstrate the effectiveness of our method even under computation-constrained scenarios, we integrate our Factorized Features framework to state-of-the-art lightweight super-resolution models and observe consistent improvement.
>
> | Method          | Latency (ms) | #FLOPs (G) | #Params (K) | Set5 PSNR / SSIM | Set14 PSNR / SSIM | B100 PSNR / SSIM | Urban100 PSNR / SSIM | Manga109 PSNR / SSIM |
> |-----------------|:------------:|:----------:|:-----------:|:----------------:|:-----------------:|:----------------:|:--------------------:|:--------------------:|
> | SwinIR-lt       | 222.9        | 63.6       | 930         | 32.44 / 0.8976   | 28.77 / 0.7858    | 27.69 / 0.7406   | 26.47 / 0.7980       | 30.92 / 0.9151       |
> | ELAN-lt         | 18.0         | 54.1       | 640         | 32.43 / 0.8975   | 28.78 / 0.7858    | 27.69 / 0.7406   | 26.54 / 0.7982       | 30.92 / 0.9150       |
> | OmniSR          | 22.5         | 50.9       | 792         | 32.49 / 0.8988   | 28.78 / 0.7859    | 27.71 / 0.7415   | 26.64 / 0.8018       | 31.02 / 0.9150       |
> | SRFormer-lt     | 287.2        | 62.8       | 873         | 32.51 / 0.8988   | 28.82 / 0.7872    | 27.73 / 0.7422   | 26.67 / 0.8032       | 31.17 / 0.9165       |
> | ATD-lt          | 189.7        | 100.1      | 769         | 32.63 / 0.8998   | 28.89 / 0.7886    | 27.79 / 0.7440   | 26.97 / 0.8107       | 31.48 / 0.9198       |
> | HiT-SRF         | 82.1         | 58.0       | 866         | 32.55 / 0.8999   | 28.87 / 0.7880    | 27.75 / 0.7432   | 26.80 / 0.8069       | 31.26 / 0.9171       |
> | ASID-D8         | 61.8         | 49.6†      | 748         | 32.57 / 0.8990   | 28.89 / 0.7898    | 27.78 / 0.7449   | 26.89 / 0.8096       | –                    |
> | MambaIR-lt      | 55.8         | 84.6       | 924         | 32.42 / 0.8977   | 28.74 / 0.7847    | 27.68 / 0.7400   | 26.52 / 0.7983       | 30.94 / 0.9135       |
> | MambaIRV2-lt    | 153.4        | 75.6       | 790         | 32.51 / 0.8992   | 28.84 / 0.7878    | 27.75 / 0.7426   | 26.82 / 0.8079       | 31.24 / 0.9182       |
> | RDN             | 66.0         | 1309.2     | 22271  | 32.47 / 0.8990   | 28.81 / 0.7871    | 27.72 / 0.7419   | 26.61 / 0.8028       | 31.00 / 0.9151       |
> | RCAN            | 52.2         | 917.6      | 15592       | 32.63 / 0.9002   | 28.87 / 0.7889    | 27.77 / 0.7436   | 26.82 / 0.8087       | 31.22 / 0.9173       |
> | CATANet[1] (CVPR'25) | 102.4      | 49.3       | 535         | 32.58 / 0.8998   | 28.90 / 0.7880    | 27.75 / 0.7427   | 26.87 / 0.8081       | 31.31 / 0.9183       |
> | ESC[2] (ICCV'25)     | 21.9       | 149.2      | 968         | 32.68 / 0.9011   | 28.93 / 0.7902    | 27.80 / 0.7447   | 27.07 / 0.8144       | 31.54 / 0.9207       |
> | CATANet-F (Ours)  | 110.3      | 56.7       | 941         | 32.69 / 0.9017   | 28.93 / 0.7906    | 27.82 / 0.7444   | 27.05 / 0.8133       | 31.62 / 0.9214       |
> | ESC-F (Ours)      | 24.6       | 73.4       | 952         | 32.74 / 0.9032   | 28.99 / 0.7931    | 27.86 / 0.7483   | 27.11 / 0.8158       | 31.78 / 0.9233       |
>
> [1] Liu, X., Liu, J., Tang, J., & Wu, G. (2025). CATANet: Efficient Content-Aware Token Aggregation for Lightweight Image Super-Resolution. In Proceedings of the Computer Vision and Pattern Recognition Conference (pp. 17902-17912).
>
> [2] Lee, D., Yun, S., & Ro, Y. (2025). Emulating Self-attention with Convolution for Efficient Image Super-Resolution. arXiv preprint arXiv:2503.06671.
>
> > **W2. Lack of Semantic Information**
>
> Good catch! Our current focus is on frequency-domain structural modeling rather than semantic understanding. However, our framework is extensible - for example, we can insert the CLIP feature into the Coefficient Backbone [3] or condition attention on human instructions [4] in the Basis Transformer for explicit guidance. We will leave this to future work.
>
> [3] Lu, Z., Xia, Q., Wang, W., & Wang, F. (2025). CLIP-aware domain-adaptive super-resolution. Multimedia Systems, 31(3), 257.
>
> [4] Conde, M. V., Geigle, G., & Timofte, R. (2024, September). Instructir: High-quality image restoration following human instructions. In European Conference on Computer Vision (pp. 1-21). Cham: Springer Nature Switzerland.
>
>
> > **W3. Scalability in Multi-Frame Compression**
>
> As shown in Figure 7, our method demonstrates robust scalability. The PSNR remains high even at M=24, with only a minor degradation of approximately 0.1 dB when increasing M from 8 to 24. This slight quality loss, resulting from more frames sharing common bases, is significantly outweighed by the twofold improvement in compression efficiency. This trade-off is more favorable than those offered by existing methods.
>
> Regarding Table 2, the high BD-Rate observed at small values of M (e.g., M=1) is attributed to the overhead from the additional basis transmission branch, as detailed in Figure 10 of the Supplement.
>
> > **W4. Backbone Architecture Dependence**
>
> We integrate our Factorized Features framework into state-of-the-art lightweight super-resolution models and witness consistent improvement as shown in the table of W1.
>
> > **Q1. Failure Cases**
>
> Failure cases typically involve extremely fine textures—details that are absent in the low-resolution (LR) input but appear in the super-resolved (SR) output. However, this has been a shared inherent limitation of super-resolution methods to date. For example, in the top row of Figure 6, some white structures are present in HR but are completely absent in LR.
>
> Note that due to rebuttal guidelines, we can not include images or external links here; visual examples of these failure cases will be provided in the revised manuscript.
>
> > **L1. Task-Specific Overfitting**
>
> We thank the reviewer for this question. We understand it to ask whether our Factorized Features overfit to a single task or dataset. In fact, they exhibit strong cross-task and cross-domain generalization. For example, the same Factorized Features learned for super-resolution transfer seamlessly to image compression (as described in Section 4.3) and other low-level vision tasks such as image restoration (see below).
>
> Even within super-resolution, a model trained exclusively on the DF2K dataset achieves excellent performance on held-out benchmarks (the lightweight SR comparison Table in Question 1 is trained with DF2K). If this does not address your concern, could you please clarify further?
>
> #### GoPro Dataset (Motion Deblurring)
> | Metric / Method  | DeblurGAN-v2 | SRN  | DMPHN | SDWNet | MPRNet | MIMO-UNet+ | DeepRFT+ | MAXIM-3S | Stripformer | MSDI-net | Restormer | NAFNet | FFTformer | GRL-B | MLWNet (CVPR'24) | MLWNet-F |
> |------------------|--------------|------|-------|--------|--------|------------|----------|----------|-------------|----------|-----------|--------|-----------|-------| ---------------- | -------- |
> | **PSNR**         | 29.55        | 30.26| 31.20 | 31.26  | 32.66  | 32.45      | 33.23    | 32.86    | 33.08       | 33.28    | 33.57     | 33.69  | 34.21     | 33.93 | 33.83            | 34.4     |
> | **SSIM**         | 0.934        | 0.934| 0.945 | 0.966  | 0.959  | 0.957      | 0.963    | 0.961    | 0.962       | 0.964    | 0.966     | 0.967  | 0.969     | 0.968 | 0.968            | 0.971    |
> #### Dense-Haze Dataset (Dehazing)
> | Metric / Method | DehazeNet | AOD-Net | MSBDN | FFA-Net | AECR-Net | DeHamer | PMNet | DehazeFormer | TaylorFormer | LH-Net | MITNet | PGH^2Net (AAAI'24) | ConvIR (TPAMI'24) | ConvIR-F|
> |-----------------|-----------|---------|-------|---------|----------|---------|-------|--------------|--------------|--------|--------|-----------         | ---               | --      |
> | **PSNR**        | 13.84     | 13.14   | 15.37 | 14.39   | 15.80    | 16.62   | 16.79 | 16.29        | 16.66        | 18.87  | 16.97  | 17.02              | 17.45             | 18.12   |
> | **SSIM**        | 0.43      | 0.41    | 0.49  | 0.45    | 0.47     | 0.56    | 0.51  | 0.51         | 0.56         | 0.561  | 0.606  | 0.61               | 0.802             | 0.823   |
>
> #### Our framework clearly generalizes across diverse low-level vision tasks. Due to time constraints in the rebuttal phase, we have focused on these two additional benchmarks, but we will include further evaluations on denoising, deraining, and other tasks in the revised manuscript.
>
> > **S&W1. Generalization**
>
> We can indeed extend our network to other tasks. For image generation, for example, we can integrate our Factorized Features into the VAE decoder of a diffusion model. For video generation, we can employ our mergeable basis and leverage its properties to enhance structural coherence. We will discuss these extensions in the Future Work / Directions section.
>
> We hope these clarifications address the reviewer's concerns. We believe the strong empirical results, unified framework, and extensibility of our approach constitute significant contributions that merit acceptance. We will incorporate all suggested improvements in the final version.

---

> ### Comment · Area_Chair_4crZ · 2025-08-05
> **Please respond to authors' rebuttal before Aug. 6 (AOE)**
>
> Dear Reviewer yir9,
>
> This is a reminder that the author-reviewer discussion period is ending soon on Aug. 6 (AOE), and you have not yet responded to authors' rebuttal. Please read authors' rebuttal as soon as possible, and engage in any necessary discussions, and consider if you would like to update your review and score. Please at least submit the Mandatory Acknowledgement as a sign that you have completed this task.
>
> Thank you for your service in the review process.
>
> AC

---

### Note · Authors · 2025-08-12

Dear AC and Reviewers,

We sincerely thank all reviewers for their insightful feedback and constructive engagement throughout the discussion phase. We deeply appreciate the recognition of our **novel frequency-aware decomposition approach (yir9, Wt2E, YrvH, evQ6)** and acknowledgment that our Factorized Features represents a **"significant and novel contribution"** that moves beyond purely architectural improvements (YrvH). We value the positive assessment of our strong empirical results achieving **204.4% PSNR improvement in SR and 9.35% BD-rate reduction in compression over baseline (all reviewers)** and recognition of our unified framework's broad generalizability across multiple low-level vision tasks (yir9, Wt2E).

We are pleased that ALL reviewers have positively responded to our paper and rebuttal with concerns resolved, where reviewer YrvH upgrades the score:

- yir9: "Interesting paper" with strong empirical performance - acknowledged all contributions.
- Wt2E: "All questions have been addressed" - appreciated well-structured response.
- YrvH: "Results have mostly addressed my concerns" - **INCREASED score after rebuttal**.
- evQ6: "All my concerns are solved" - confirmed positive engagement.


During the rebuttal, we provided substantial new evidence:
- Lightweight model experiments (CATANet-F, ESC-F) demonstrate the proposed method achieves consistent gains under computational constraints.
- Extended task evaluations on deblurring (GoPro) and dehazing (Dense-Haze) confirm the broad generalization of our approach.
- Comprehensive ablation studies on coordinate transformation functions as well as $N$ and $\alpha$ parameters are conducted as requested.
- Technical clarifications on formulations, learned basis mechanism, and mergeable-basis property are provided.

We commit to incorporating all suggested improvements in the final version:

- Enhancing Figure 5 captions and improved readability of Figures.
- Clarifying notations, visualization, and explanation.
- Integrating the requested lightweight comparisons and additional task results.
- Including potential failure cases.

We believe our Factorized Features framework makes a genuine contribution by introducing a principled frequency-domain representation that achieves SOTA performance across both super-resolution and compression while demonstrating extensibility to other low-level vision tasks.

Thank you again for your valuable time and insights in refining this work.

Best regards,

Authors

---

### Decision · Program_Chairs · 2025-09-17

**Decision:**

Accept (poster)

**Comment:**

This submission presents Factorized Features, which represents an image by learning to produce both the expansion coefficient features and basis functions. With this representation, the work developed techniques for low-level vision tasks, particularly single image super-resolution and image compression, and empirical performance is shown to surpass existing approaches.

Reviewers and I appreciate the novelty of this representation framework, and the impressive empirical strength. Reviewers mentioned a few insufficiencies of the methods, primarily in the limited low-level vision task demonstration and efficiency issues, for which the authors provided more experimental results on deblurring and dehazing tasks and with light weighted models, with supportive performance. The authors also provided further clarifications of technical choices and result explanations. Overall, this paper constitutes a nice contribution to the community. I hope the authors to include the additional results and further clarifications into the final paper.